# Robust Deep Reinforcement Learning against Adversarial Perturbations on State Observations

**Huan Zhang**[*,1]    **Hongge Chen**[*,2]    **Chaowei Xiao**[3]
**Bo Li**[4]    **Mingyan Liu**[5]    **Duane Boning**[2]    **Cho-Jui Hsieh**[1]
[1]UCLA    [2]MIT    [3]NVIDIA    [4]UIUC    [5]University of Michigan
huan@huan-zhang.com, chenhg@mit.edu, chaoweix@nvidia.com,
lbo@illinois.edu,mingyan@umich.edu,boning@mtl.mit.edu,chohsieh@cs.ucla.edu
[*]Huan Zhang and Hongge Chen contributed equally.

## Abstract

A deep reinforcement learning (DRL) agent observes its states through observations, which may contain natural measurement errors or adversarial noises. Since the observations deviate from the true states, they can mislead the agent into making suboptimal actions. Several works have shown this vulnerability via adversarial attacks, but existing approaches on improving the robustness of DRL under this setting have limited success and lack for theoretical principles. We show that naively applying existing techniques on improving robustness for classification tasks, like adversarial training, are ineffective for many RL tasks. We propose the state-adversarial Markov decision process (SA-MDP) to study the fundamental properties of this problem, and develop a theoretically principled policy regularization which can be applied to a large family of DRL algorithms, including proximal policy optimization (PPO), deep deterministic policy gradient (DDPG) and deep Q networks (DQN), for both discrete and continuous action control problems. We significantly improve the robustness of PPO, DDPG and DQN agents under a suite of strong white box adversarial attacks, including new attacks of our own. Additionally, we find that a robust policy noticeably improves DRL performance even without an adversary in a number of environments. Our code is available at https://github.com/chenhongge/StateAdvDRL.

## 1   Introduction

With deep neural networks (DNNs) as powerful function approximators, deep reinforcement learning (DRL) has achieved great success on many complex tasks [46, 35, 33, 64, 20] and even on some safety-critical applications (e.g., autonomous driving [74, 56, 49]). Despite achieving super-human level performance on many tasks, the existence of adversarial examples [69] in DNNs and many successful attacks to DRL [27, 4, 36, 50, 81] motivates us to study robust DRL algorithms.

When an RL agent obtains its current state via observations, the observations may contain uncertainty that naturally originates from unavoidable sensor errors or equipment inaccuracy. A policy not robust to such uncertainty can lead to catastrophic failures (e.g., the navigation setting in Figure 1). To ensure safety under the *worst case* uncertainty, we consider the adversarial setting where the state observation is adversarially perturbed from $s$ to $\nu(s)$, yet the underlying true environment state $s$ is unchanged. This setting is aligned with many adversarial attacks on state observations (e.g., [27, 36]) and cannot be characterized by existing tools such as partially observable Markov decision process (POMDP), because the conditional observation probabilities in POMDP cannot capture the adversarial (worst case) scenario. Studying the fundamental principles in this setting is crucial.

Before basic principles were developed, several early approaches [5, 40, 50] extended existing adversarial defenses for supervised learning, e.g., adversarial training [32, 39, 87] to improve robustness

under this setting. Specifically, we can attack the agent and generate trajectories adversarially during training time, and apply any existing DRL algorithm to hopefully obtain a robust policy. Unfortunately, we show that for most environments, naive adversarial training (e.g., putting adversarial states into the replay buffer) leads to unstable training and deteriorates agent performance [5, 15], or does not significantly improve robustness under strong attacks. Since RL and supervised learning are quite different problems, naively applying techniques from supervised learning to RL without a proper theoretical justification can be unsuccessful. To summarize, we study the theory and practice of robust RL against perturbations on state observations:

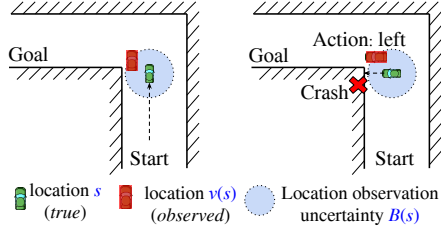

Figure 1: A car observes its location through sensors (e.g., GPS) and plans its route to the goal. Without considering the uncertainty in observed location (e.g., error of GPS coordinates), an unsafe policy may crash into the wall because $s \neq \nu(s)$.

• We formulate the perturbation on state observations as a modified Markov decision process (MDP), which we call state-adversarial MDP (SA-MDP), and study its fundamental properties. We show that under an optimal adversary, a stationary and Markovian optimal policy may not exist for SA-MDP.

• Based on our theory of SA-MDP, we propose a theoretically principled robust policy regularizer which is related to the total variation distance or KL-divergence on perturbed policies. It can be practically and efficiently applied to a wide range of RL algorithms, including PPO, DDPG and DQN.

• We conduct experiments on 10 environments ranging from Atari games with discrete actions to complex control tasks in continuous action space. Our proposed method significantly improves robustness under strong white-box attacks on state observations, including two *strong* attacks we design, the robust Sarsa attack (RS attack) and maximal action difference attack (MAD attack).

## 2 Related Work

**Robust Reinforcement Learning** Since each element of RL (observations, actions, transition dynamics and rewards) can contain uncertainty, robust RL has been studied from different perspectives. Robust Markov decision process (RMDP) [29, 47] considers the worst case perturbation from transition probabilities, and has been extended to distributional settings [82] and partially observed MDPs [48]. The agent observes the original true state from the environment and acts accordingly, but the environment can choose from a set of transition probabilities that minimizes rewards. Compared to our SA-MDP where the adversary changes only observations, in RMDP the ground-truth states are changed so RMDP is more suitable for modeling *environment parameter changes* (e.g., changes in physical parameters like mass and length, etc). RMDP theory has inspired robust deep Q-learning [62] and policy gradient algorithms [41, 12, 42] that are robust against small environmental changes.

Another line of works [51, 34] consider the adversarial setting of multi-agent reinforcement learning [70, 9]. In the simplest two-player setting (referred to as minimax games [37]), each agent chooses an action at each step, and the environment transits based on both actions. The regular $Q$ function $Q(s, a)$ can be extended to $Q(S, a, o)$ where $o$ is the opponent's action and Q-learning is still convergent. This setting can be extended to deep Q learning and policy gradient algorithms [34, 51]. Pinto et al. [51] show that learning an opponent simultaneously can improve the agent's performance as well as its robustness against environment turbulence and test conditions (e.g., change in mass or friction). Gu et al. [21] carried out real-world experiments on the two-player adversarial learning game. Tessler et al. [71] considered adversarial perturbations on the action space. Fu et al. [16] investigated how to learn a robust reward. All these settings are different from ours: we manipulate only the state observations but do not change the underlying environment (the true states) directly.

**Adversarial Attacks on State Observations in DRL** Huang et al. [27] evaluated the robustness of deep reinforcement learning policies through an FGSM based attack on Atari games with discrete actions. Kos & Song [31] proposed to use the value function to guide adversarial perturbation search. Lin et al. [36] considered a more complicated case where the adversary is allowed to attack only a subset of time steps, and used a generative model to generate attack plans luring the agent to a designated target state. Behzdan & Munir [4] studied black-box attacks on DQNs with discrete actions via transferability of adversarial examples. Pattanaik et al. [50] further enhanced adversarial attacks to DRL with multi-step gradient descent and better engineered loss functions. They require a critic or $Q$ function to perform attacks. Typically, the critic learned during agent training is used.

We find that using this critic can be sub-optimal or impractical in many cases, and propose our two *critic-independent* and strong attacks (RS and MAD attacks) in Section 3.5. We refer the reader to recent surveys [81, 28] for a taxonomy and a comprehensive list of adversarial attacks in DRL setting.

**Improving Robustness for State Observations in DRL**  For discrete action RL tasks, Kos & Song [31] first presented preliminary results of adversarial training on Pong (one of the simplest Atari environments) using weak FGSM attacks on pixel space. Behzadan & Munir [5] applied adversarial training to several Atari games with DQN, and found it challenging for the agent to adapt to the attacks during training time. These early approaches achieved much worse results than ours: for Pong, Behzadan & Munir [5] can improve reward under attack from $-21$ (lowest) to $-5$, yet is still far away from the optimal reward ($+21$). Recently, Mirman et al. [43], Fischer et al. [15] treat the *discrete action* outputs of DQN as labels, and apply existing certified defense for classification [44] to robustly predict actions using imitation learning. This approach outperforms [5], but it is unclear how to apply it to environments with continuous action spaces. Compared to their approach, our SA-DQN does not use imitation learning and achieves better performance on most environments.

For continuous action RL tasks (e.g., MuJoCo environments in OpenAI Gym), Mandlekar et al. [40] used a weak FGSM based attack with policy gradient to adversarially train a few simple RL tasks. Pattanaik et al. [50] used stronger multi-step gradient based attacks; however, their evaluation focused on robustness against environmental changes rather than state perturbations. Unlike our work which first develops principles and then applies to different DRL algorithms, these works directly extend adversarial training in supervised learning to the DRL setting and do not reliably improve test time performance under strong attacks in Section 4. A concurrent work [63] considers a smoothness regularizer similar to ours, but they use virtual adversarial training and focus on improving generalization instead of robustness. We provide theoretical justifications for our regularizer, propose new attacks and conduct comprehensive empirical evaluations under strong adversaries.

Other related works include [24], which proposed a meta online learning procedure with a master agent detecting the presence of the adversary and switching between a few sub-policies, but did not discuss how to train a single agent robustly. [11] applied adversarial training specifically for RL-based path-finding algorithms. [38] considered the worst-case scenario during rollouts for existing DQN agents to ensure safety, but it relies on an existing policy and does not include a training procedure.

# 3 Methodology

## 3.1 State-Adversarial Markov Decision Process (SA-MDP)

**Notations**  A Markov decision process (MDP) is defined as $(\mathcal{S}, \mathcal{A}, R, p, \gamma)$, where $\mathcal{S}$ is the state space, $\mathcal{A}$ is the action space, $R : \mathcal{S} \times \mathcal{A} \times \mathcal{S} \to \mathbb{R}$ is the reward function, and $p : \mathcal{S} \times \mathcal{A} \to \mathcal{P}(\mathcal{S})$ is the transition probability of environment, where $\mathcal{P}(\mathcal{S})$ defines the set of all possible probability measures on $\mathcal{S}$. The transition probability $p(s'|s, a) = \Pr(s_{t+1} = s'|s_t = s, a_t = a)$, where $t$ is the time step. We denote a stationary policy as $\pi : \mathcal{S} \to \mathcal{P}(\mathcal{A})$, the set of all stochastic and Markovian policies as $\Pi_{\mathrm{MR}}$, the set of all deterministic and Markovian policies as $\Pi_{\mathrm{MD}}$. Discount factor $0 < \gamma < 1$.

In state-adversarial MDP (SA-MDP), we introduce an adversary $\nu(s) : \mathcal{S} \to \mathcal{S}$ [1]. The adversary perturbs only the state observations of the agent, such that the action is taken as $\pi(a|\nu(s))$; the environment still transits from the true state $s$ rather than $\nu(s)$ to the next state. Since $\nu(s)$ can be different from $s$, the agent's action from $\pi(a|\nu(s))$ may be sub-optimal, and thus the adversary is able to reduce the reward. In real world RL problems, the adversary can be reflected as the worst case noise in measurement or state estimation uncertainty. Note that this scenario is different from the two-player Markov game [37] where both players see unperturbed true environment states and interact with the environment directly; the opponent's action can change the true state of the game.

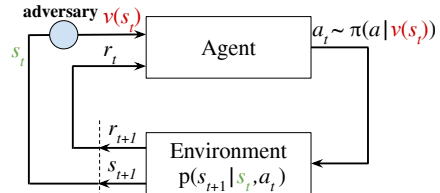

Figure 2: Reinforcement learning with perturbed state observations. The agent observes a perturbed state $\nu(s_t)$ rather than the true environment state $s_t$.

To allow a formal analysis, we first make the assumption for the adversary $\nu$:

**Assumption 1** (Stationary, Deterministic and Markovian Adversary). *$\nu(s)$ is a deterministic function $\nu : \mathcal{S} \to \mathcal{S}$ which only depends on the current state $s$, and $\nu$ does not change over time.*

This assumption holds for many adversarial attacks [27, 36, 31, 50]. These attacks only depend on the current state input and the policy or Q network so they are Markovian; the network parameters are frozen at test time, so given the same $s$ the adversary will generate the same (stationary) perturbation. We leave the formal analysis of non-Markovian, non-stationary adversaries as future work.

If the adversary can perturb a state $s$ arbitrarily without bounds, the problem can become trivial. To fit our analysis to the most realistic settings, we need to restrict the power of an adversary. We define perturbation set $B(s)$, to restrict the adversary to perturb a state $s$ only to a predefined set of states:

**Definition 1** (Adversary Perturbation Set). *We define a set $B(s)$ which contains all allowed perturbations of the adversary. Formally, $\nu(s) \in B(s)$ where $B(s)$ is a set of states and $s \in \mathcal{S}$.*

$B(s)$ is usually a set of task-specific "neighboring" states of $s$ (e.g., bounded sensor measurement errors), which makes the observation still meaningful (yet not accurate) even with perturbations. After defining $B$, an SA-MDP can be represented as a 6-tuple $(\mathcal{S}, \mathcal{A}, B, R, p, \gamma)$.

**Analysis of SA-MDP** We first derive Bellman Equations and a basic policy evaluation procedure, then we discuss the possibility of obtaining an optimal policy for SA-MDP. The adversarial value and action-value functions under $\nu$ in an SA-MDP are similar to those of a regular MDP:

$$\tilde{V}_{\pi \circ \nu}(s) = \mathbb{E}_{\pi \circ \nu}\left[\sum_{k=0}^{\infty} \gamma^k r_{t+k+1} | s_t = s\right], \quad \tilde{Q}_{\pi \circ \nu}(s,a) = \mathbb{E}_{\pi \circ \nu}\left[\sum_{k=0}^{\infty} \gamma^k r_{t+k+1} | s_t = s, a_t = a\right],$$

where the reward at step-$t$ is defined as $r_t$ and $\pi \circ \nu$ denotes the policy under observation perturbations: $\pi(a|\nu(s))$. Based on these two definitions, we first consider the simplest case with *fixed* $\pi$ and $\nu$:

**Theorem 1** (Bellman equations for fixed $\pi$ and $\nu$). *Given $\pi : \mathcal{S} \to \mathcal{P}(\mathcal{A})$ and $\nu : \mathcal{S} \to \mathcal{S}$, we have*

$$\tilde{V}_{\pi \circ \nu}(s) = \sum_{a \in \mathcal{A}} \pi(a|\nu(s)) \sum_{s' \in \mathcal{S}} p(s'|s,a)\left[R(s,a,s') + \gamma \tilde{V}_{\pi \circ \nu}(s')\right]$$

$$\tilde{Q}_{\pi \circ \nu}(s,a) = \sum_{s' \in \mathcal{S}} p(s'|s,a)\left[R(s,a,s') + \gamma \sum_{a' \in \mathcal{A}} \pi(a'|\nu(s'))\tilde{Q}_{\pi \circ \nu}(s',a')\right].$$

The proof of Theorem 1 is simple, as when $\pi, \nu$ are fixed, they can be "merged" as a single policy, and existing results from MDP can be directly applied. Now we consider a more complicated case, where we want to find the value functions under *optimal adversary* $\nu^*(\pi)$, minimizing the total expected reward for a *fixed* $\pi$. The optimal adversarial value and action-value functions are defined as:

$$\tilde{V}_{\pi \circ \nu^*}(s) = \min_{\nu} \tilde{V}_{\pi \circ \nu}(s), \quad \tilde{Q}_{\pi \circ \nu^*}(s,a) = \min_{\nu} \tilde{Q}_{\pi \circ \nu}(s,a).$$

**Theorem 2** (Bellman contraction for optimal adversary). *Define Bellman operator $\mathscr{L} : \mathbb{R}^{|\mathcal{S}|} \to \mathbb{R}^{|\mathcal{S}|}$,*

$$(\mathscr{L}\tilde{V})(s) = \min_{s_{\nu} \in B(s)} \sum_{a \in \mathcal{A}} \pi(a|s_{\nu}) \sum_{s' \in \mathcal{S}} p(s'|s,a)\left[R(s,a,s') + \gamma \tilde{V}(s')\right]. \quad (1)$$

*The Bellman equation for optimal adversary $\nu^*$ can then be written as: $\tilde{V}_{\pi \circ \nu^*} = \mathscr{L}\tilde{V}_{\pi \circ \nu^*}$. Additionally, $\mathscr{L}$ is a contraction that converges to $\tilde{V}_{\pi \circ \nu^*}$.*

Theorem 2 says that given a *fixed* policy $\pi$, we can evaluate its performance (value functions) under the optimal (strongest) adversary, through a Bellman contraction. It is functionally similar to the "policy evaluation" procedure in regular MDP. The proof of Theorem 2 is in the same spirit as the proof of Bellman optimality equations for solving the optimal policy for an MDP; the important difference here is that we solve the optimal adversary, for a *fixed* policy $\pi$. Given $\pi$, value functions for MDP and SA-MDP can be vastly different. Here we show a 3-state toy environment in Figure 3; an optimal MDP policy is to take action 2 in $S_1$, action 1 in $S_2$ and $S_3$. Under the presence of an adversary $\nu(S_1) = S_2$, $\nu(S_2) = S_1$, $\nu(S_3) = S_1$, this policy receives zero total reward as the adversary can make the action $\pi(a|\nu(s))$ totally wrong regardless of the states. On the other hand, a

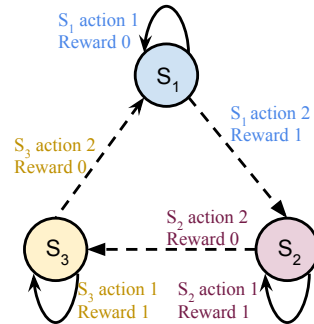

Figure 3: A toy environment.

policy taking random actions on all three states (which is a non-optimal policy for MDP) is unaffected by the adversary and obtains non-zero rewards in SA-MDP. Details are given in Appendix A.

Finally, we discuss our ultimate quest of finding an *optimal* policy $\pi^*$ under the strongest adversary $\nu^*(\pi)$ in the SA-MDP setting (we use the notation $\nu^*(\pi)$ to explicit indicate that $\nu^*$ is the optimal

adversary for a given $\pi$). An optimal policy should be the best among all policies on every state:

$$\tilde{V}_{\pi^* \circ \nu^*(\pi^*)}(s) \geq \tilde{V}_{\pi \circ \nu^*(\pi)}(s) \quad \text{for } \forall s \in \mathcal{S} \text{ and } \forall \pi, \tag{2}$$

where both $\pi$ and $\nu$ are not fixed. The first question is, what policy classes we need to consider for $\pi^*$. In MDPs, deterministic policies are sufficient. We show that this does not hold anymore in SA-MDP:

**Theorem 3.** *There exists an SA-MDP and some stochastic policy $\pi \in \Pi_{MR}$ such that we cannot find a better deterministic policy $\pi' \in \Pi_{MD}$ satisfying $\tilde{V}_{\pi' \circ \nu^*(\pi')}(s) \geq \tilde{V}_{\pi \circ \nu^*(\pi)}(s)$ for all $s \in \mathcal{S}$.*

The proof is done by constructing a counterexample where some stochastic policies are better than *any* other deterministic policies in SA-MDP (see Appendix A). Contrarily, in MDP, for any stochastic policy we can find a deterministic policy that is at least as good as the stochastic one. Unfortunately, even looking for both deterministic and stochastic policies still cannot always find an optimal one:

**Theorem 4.** *Under the optimal $\nu^*$, an optimal policy $\pi^* \in \Pi_{MR}$ does not always exist for SA-MDP.*

The proof follows the same counterexample as in Theorem 3. The optimal policy $\pi^*$ requires to have $\tilde{V}_{\pi^* \circ \nu^*(\pi^*)}(s) \geq \tilde{V}_{\pi \circ \nu^*(\pi)}(s)$ for all $s$ and any $\pi$. In an SA-MDP, sometimes we have to make a trade-off between the value of states and no policy can maximize the values of all states.

Despite the difficulty of finding an optimal policy under the optimal adversary, we show that under certain assumptions, the loss in performance due to an optimal adversary can be bounded:

**Theorem 5.** *Given a policy $\pi$ for a non-adversarial MDP and its value function is $V_\pi(s)$. Under the optimal adversary $\nu$ in SA-MDP, for all $s \in \mathcal{S}$ we have*

$$\max_{s \in \mathcal{S}} \left\{ V_\pi(s) - \tilde{V}_{\pi \circ \nu^*(\pi)}(s) \right\} \leq \alpha \max_{s \in \mathcal{S}} \max_{\hat{s} \in B(s)} \mathrm{D}_{\mathrm{TV}}(\pi(\cdot|s), \pi(\cdot|\hat{s})) \tag{3}$$

*where $\mathrm{D}_{\mathrm{TV}}(\pi(\cdot|s), \pi(\cdot|\hat{s}))$ is the total variation distance between $\pi(\cdot|s)$ and $\pi(\cdot|\hat{s})$, and $\alpha := 2[1 + \frac{\gamma}{(1-\gamma)^2}] \max_{(s,a,s') \in \mathcal{S} \times \mathcal{A} \times \mathcal{S}} |R(s, a, s')|$ is a constant that does not depend on $\pi$.*

Theorem 5 says that as long as differences between the action distributions under state perturbations (the term $\mathrm{D}_{\mathrm{TV}}(\pi(\cdot|s), \pi(\cdot|\hat{s}))$) are not too large, the performance gap between $\tilde{V}_{\pi \circ \nu^*}(s)$ (state value of SA-MDP) and $V_\pi(s)$ (state value of regular MDP) can be bounded. An important consequence is the motivation of regularizing $\mathrm{D}_{\mathrm{TV}}(\pi(\cdot|s), \pi(\cdot|\hat{s}))$ during training to obtain a policy robust to strong adversaries. The proof is based on tools developed in constrained policy optimization [1], which gives an upper bound on value functions given two policies with bounded divergence. In our case, we desire that a bounded state perturbation $\hat{s}$ produces bounded divergence between $\pi(\cdot|s)$ and $\pi(\cdot|\hat{s})$.

We now study a few practical DRL algorithms, including both deep Q-learning (DQN) for discrete actions and actor-critic based policy gradient methods (DDPG and PPO) for continuous actions.

## 3.2 State-Adversarial DRL for Stochastic Policies: A Case Study on PPO

We start with the most general case where the policy $\pi(a|s)$ is stochastic (e.g., in PPO [59]). The total variation distance is not easy to compute for most distributions, so we upper bound it again by KL divergence: $\mathrm{D}_{\mathrm{TV}}(\pi(a|s), \pi(a|\hat{s})) \leq \sqrt{\frac{1}{2}\mathrm{D}_{\mathrm{KL}}(\pi(a|s)\|\pi(a|\hat{s}))}$. When Gaussian policies are used, we denote $\pi(a|s) \sim \mathcal{N}(\mu_s, \Sigma_s)$ and $\pi(a|\hat{s}) \sim \mathcal{N}(\mu_{\hat{s}}, \Sigma_{\hat{s}})$. The KL-divergence can be given as:

$$\mathrm{D}_{\mathrm{KL}}(\pi(a|s)\|\pi(a|\hat{s})) = \frac{1}{2} \left( \log |\Sigma_{\hat{s}} \Sigma_s^{-1}| + \mathrm{tr}(\Sigma_{\hat{s}}^{-1} \Sigma_s) + (\mu_{\hat{s}} - \mu_s)^\top \Sigma_{\hat{s}}^{-1}(\mu_{\hat{s}} - \mu_s) - |\mathcal{A}| \right). \tag{4}$$

Regularizing KL distance (4) for all $\hat{s} \in B(s)$ will lead to a smaller upper bound in (21), which is directly related to agent performance under optimal adversary. In PPO, the mean terms $\mu_s, \mu_{\hat{s}}$ are produced by neural networks: $\mu_{\theta_\mu}(s)$ and $\mu_{\theta_\mu}(\hat{s})$, and we assume $\Sigma$ is a diagonal matrix independent of state $s$ ($\Sigma_{\hat{s}} = \Sigma_s = \Sigma$). Regularizing the above KL-divergence over all $s$ from sampled trajectories and all $\hat{s} \in B(s)$ leads to the following state-adversarial regularizer for PPO, ignoring constant terms:

$$\mathcal{R}_{\mathrm{PPO}}(\theta_\mu) = \frac{1}{2} \sum_s \max_{\hat{s} \in B(s)} \left( \mu_{\theta_\mu}(\hat{s}) - \mu_{\theta_\mu}(s) \right)^\top \Sigma^{-1} \left( \mu_{\theta_\mu}(\hat{s}) - \mu_{\theta_\mu}(s) \right) := \frac{1}{2} \sum_s \max_{\hat{s} \in B(s)} \mathcal{R}_s(\hat{s}, \theta_\mu). \tag{5}$$

We replace $\max_{s \in \mathcal{S}}$ term in Theorem 5 with a more practical and optimizer-friendly summation over all states in sampled trajectory. A similar treatment was used in TRPO [33] which was also derived as a KL-based regularizer, albeit on $\theta_\mu$ space rather than on state space. However, minimizing (5) is challenging as it is a minimax objective, and we also have $\nabla_{\hat{s}} \mathcal{R}(\hat{s}, \theta_\mu)|_{\hat{s}=s} = 0$ so using gradient

descent directly cannot solve the inner maximization problem to a local maximum. Instead of using the more expensive second order methods, we propose two first order approaches to solve (5): convex relaxations of neural networks, and Stochastic Gradient Langevin Dynamics (SGLD). Here we focus on discussing convex relaxation based method, and we defer SGLD based solver to Section C.2.

Convex relaxation of non-linear units in neural networks enables an efficient analysis of the outer bounds for a neural network [79, 86, 66, 13, 78, 76, 57, 67]. Several works have used it for certified adversarial defenses [80, 44, 75, 19, 88], but here we leverage it as a generic optimization tool for solving minimax functions involving neural networks. Using this technique, we can obtain an upper bound for $\mathcal{R}_s(\hat{s}, \theta_\mu)$: $\overline{\mathcal{R}}_s(\theta_\mu) \geq \mathcal{R}_s(\hat{s}, \theta_\mu)$ for all $\hat{s} \in B(s)$. $\overline{\mathcal{R}}_s(\theta_\mu)$ is also a function of $\theta_\mu$ and can be seen as a transformed neural network (e.g., the dual network in Wong & Kolter [79]), and computing $\overline{\mathcal{R}}_s(\theta_\mu)$ is only a constant factor slower than computing $\mathcal{R}_s(s, \theta_\mu)$ (for a fixed $s$) when an efficient relaxation [44, 19, 88] is used. We can then solve the following minimization problem:

$$\min_{\theta_\mu} \frac{1}{2} \sum_s \overline{\mathcal{R}}_s(\theta_\mu) \geq \min_{\theta_\mu} \frac{1}{2} \sum_s \max_{\hat{s} \in B(s)} \mathcal{R}_s(\hat{s}, \theta_\mu) = \min_{\theta_\mu} \mathcal{R}_{\text{PPO}}(\theta_\mu).$$

Since we minimize an *upper bound* of the inner max, the original objective (5) is guaranteed to be minimized. Using convex relaxations can also provide certain *robustness certificates* for DRL as a bonus (e.g., we can guarantee an action has bounded changes under bounded perturbations), discussed in Appendix E. We use `auto_LiRPA`, a recently developed tool [83], to give $\overline{\mathcal{R}}_s(\theta_\mu)$ efficiently and automatically. Once the inner maximization problem is solved, we can add $\mathcal{R}_{\text{PPO}}$ as part of the policy optimization objective, and solve PPO using stochastic gradient descent (SGD) as usual.

Although Eq (5) looks similar to smoothness based regularizers in (semi-)supervised learning settings to avoid overfitting [45] and improve robustness [87], our regularizer is based on the foundations of SA-MDP. Our theory justifies the use of such a regularizer in reinforcement learning setting, while [45, 87] are developed for quite different settings not related to reinforcement learning.

### 3.3 State-Adversarial DRL for Deterministic Policies: A Case Study on DDPG

DDPG learns a deterministic policy $\pi(s) : \mathcal{S} \to \mathcal{A}$, and in this situation, the total variation distance $D_{TV}(\pi(\cdot|s), \pi(\cdot|\hat{s}))$ is malformed, as the densities at different states $s$ and $\hat{s}$ are very likely to be completely non-overlapping. To address this issue, we define a smoothed version of policy, $\bar{\pi}(a|s)$ in DDPG, where we add independent Gaussian noise with variance $\sigma^2$ to each action: $\bar{\pi}(a|s) \sim \mathcal{N}(\pi(s), \sigma^2 I_{|\mathcal{A}|})$. Then we can compute $D_{TV}(\bar{\pi}(\cdot|s), \bar{\pi}(\cdot|\hat{s}))$ using the following theorem:

**Theorem 6.** $D_{TV}(\bar{\pi}(\cdot|s), \bar{\pi}(\cdot|\hat{s})) = \sqrt{2/\pi}\frac{d}{\sigma} + O(d^3)$, *where* $d = \|\pi(s) - \pi(\hat{s})\|_2$.

Thus, as long as we can penalize $\sqrt{2/\pi}\frac{d}{\sigma}$, the total variation distance between the two smoothed distributions can be bounded. In DDPG, we parameterize the policy as a policy network $\pi_{\theta_\pi}$. Based on Theorem 5, the robust policy regularizer for DDPG is:

$$\mathcal{R}_{\text{DDPG}}(\theta_\pi) = \sqrt{2/\pi}(1/\sigma) \sum_s \max_{\hat{s} \in B(s)} \|\pi_{\theta_\pi}(s) - \pi_{\theta_\pi}(\hat{s})\|_2 \tag{6}$$

for each state $s$ in a sampled batch of states, we need to solve a maximization problem, which can be done using SGLD or convex relaxations similarly as we have shown in Section 3.2. Note that the smoothing procedure can be done completely at test time, and during training time our goal is to keep $\max_{\hat{s} \in B(s)} \|\pi_{\theta_\pi}(s) - \pi_{\theta_\pi}(\hat{s})\|_2$ small. We show the full SA-DDPG algorithm in Appendix G.

### 3.4 State-Adversarial DRL for Q Learning: A Case Study on DQN

The action space for DQN is finite, and the deterministic action is determined by the max $Q$ value: $\pi(a|s) = 1$ when $a = \arg\max_{a'} Q(s, a')$ and 0 otherwise. The total variation distance in this case is

$$D_{TV}(\pi(\cdot|s), \pi(\cdot|\hat{s})) = \begin{cases} 0 & \arg\max_a \pi(a|s) = \arg\max_a \pi(a|\hat{s}) \\ 1 & \text{otherwise.} \end{cases}$$

Thus, we want to make the top-1 action stay unchanged after perturbation, and we can use a hinge-like robust policy regularizer, where $a^*(s) = \arg\max_a Q_\theta(s, a)$ and $c$ is a small positive constant:

$$\mathcal{R}_{\text{DQN}}(\theta) := \sum_s \max\{\max_{\hat{s} \in B(s)} \max_{a \neq a^*} Q_\theta(\hat{s}, a) - Q_\theta(\hat{s}, a^*(s)), -c\}. \tag{7}$$

The sum is over all $s$ in a sampled batch. Other loss functions (e.g., cross-entropy) are also possible as long as the aim is to keep the top-1 action to stay unchanged after perturbation. This setting is

similar to the robustness of classification tasks, if we treat $a^*(s)$ as the "correct" label, thus many robust classification techniques can be applied as in [43, 15]. The maximization can be solved using projected gradient descent (PGD) or convex relaxation of neural networks. Due to its similarity to classification, we defer the details on solving $\mathcal{R}_{\text{DQN}}(\theta)$ and full SA-DQN algorithm to Appendix H.

## 3.5 Robust Sarsa (RS) and Maximal Action Difference (MAD) Attacks

In this section we propose two strong adversarial attacks under Assumption 1 for continuous action tasks trained using PPO or DDPG. For this setting, Pattanaik et al. [50] and many follow-on works use the gradient of $Q(s, a)$ to provide the direction to update states adversarially in $K$ steps:

$$s^{k+1} = s^k - \eta \cdot \text{proj}\left[\nabla_{s^k} Q(s^0, \pi(s^k))\right], \quad k = 0, \ldots, K-1, \text{and define } \hat{s} := s^K. \quad (8)$$

Here proj$[\cdot]$ is a projection to $B(s)$, $\eta$ is the learning rate, and $s^0$ is the state under attack. It attempts to find a state $\hat{s}$ triggering an action $\pi(\hat{s})$ minimizing the action-value at state $s^0$. The formulation in [50] has a glitch that the gradient is evaluated as $\nabla_{s^k} Q(s^k, \pi(s^k))$ rather than $\nabla_{s^k} Q(s^0, \pi(s^k))$. We found that the corrected form (8) is more successful. If $Q$ is a perfect action-value function, $\hat{s}$ leads to the worst action that minimizes the value at $s^0$. However, this attack has a few drawbacks:

- Attack strength strongly depends on critic quality; if $Q$ is poorly learned, is not robust against small perturbations or has obfuscated gradients, the attack fails as no correct update direction is given.
- It relies on the $Q$ function which is specific to the training process, but not used during roll-out.
- Not applicable to many actor-critic methods (e.g., TRPO and PPO) using a learned value function $V(s)$ instead of $Q(s, a)$. Finding $\hat{s} \in B(s)$ minimizing $V(s)$ does not correctly reflect the setting of perturbing observations, as $V(\hat{s})$ represents the value of $\hat{s}$ rather than the value of taking $\pi(\hat{s})$ at $s^0$.

When we evaluate the robustness of a policy, we desire it to be independent of a specific critic network to avoid these problems. We thus propose two novel *critic independent* attacks for DDPG and PPO.

**Robust Sarsa (RS) attack.** Since $\pi$ is fixed during evaluation, we can learn its corresponding $Q^\pi(s, a)$ using on-policy temporal-difference (TD) algorithms similar to Sarsa [55] without knowing the critic network used during training. Additionally, we find that the robustness of $Q^\pi(s, a)$ is very important; if $Q^\pi(s, a)$ is not robust against small perturbations (e.g., given a state $s_0$, a small change in $a$ will significantly reduce $Q^\pi(s_0, a)$ which does not reflect the true action-value), it cannot provide a good direction for attacks. Based on these, we learn $Q^\pi(s, a)$ (parameterized as an NN with parameters $\theta$) with a TD loss as in Sarsa and an additional robustness objective to minimize:

$$L_{RS}(\theta) = \sum_{i \in [N]} \left[r_i + \gamma Q_{RS}^\pi(s_i', a_i') - Q_{RS}^\pi(s_i, a_i)\right]^2 + \lambda_{RS} \sum_{i \in [N]} \max_{\hat{a} \in B(a_i)} (Q_{RS}^\pi(s_i, \hat{a}) - Q_{RS}^\pi(s_i, a_i))^2$$

$N$ is the batch size and each batch contains $N$ tuples of transitions $(s, a, r, s', a')$ sampled from agent rollouts. The first summation is the TD-loss and the second summation is the robustness penalty with regularization $\lambda_{RS}$. $B(a_i)$ is a small set near action $a_i$ (e.g., a $\ell_\infty$ ball of norm 0.05 when action is normalized between 0 to 1). The inner maximization can be solved using convex relaxation of neural networks as we have done in Section 3.3. Then, we use $Q_{\theta_{RS}}^\pi$ to perform critic-based attacks as in (8). This attack sometimes significantly outperforms the attack using the critic trained along with the policy network, as its attack strength does not depend on the quality of an existing critic. We give the detailed procedure for RS attack and show the importance of the robust objective in appendix D.

**Maximal Action Difference (MAD) attack.** We propose another simple yet very effective attack which does not depend on a critic. Following our Theorem 5 and 6, we can find an adversarial state $\hat{s}$ by *maximizing* $D_{\text{KL}}\left(\pi(\cdot|s)\|\pi(\cdot|\hat{s})\right)$. For actions parameterized by Gaussian mean $\pi_{\theta_\pi}(s)$ and covariance matrix $\Sigma$ (independent of $s$), we minimize $L_{\text{MAD}}(\hat{s}) := -D_{\text{KL}}\left(\pi(\cdot|s)\|\pi(\cdot|\hat{s})\right)$ to find $\hat{s}$:

$$\arg\min_{\hat{s} \in B(s)} L_{\text{MAD}}(\hat{s}) = \arg\max_{\hat{s} \in B(s)} \left(\pi_{\theta_\pi}(s) - \pi_{\theta_\pi}(\hat{s})\right)^\top \Sigma^{-1} \left(\pi_{\theta_\pi}(s) - \pi_{\theta_\pi}(\hat{s})\right). \quad (9)$$

For DDPG we can simply set $\Sigma = I$. The objective can be optimized using SGLD to find a good $\hat{s}$.

## 4 Experiments

In our experiments[2], the set of adversarial states $B(s)$ is defined as an $\ell_\infty$ norm ball around $s$ with a radius $\epsilon$: $B(s) := \{\hat{s} : \|s - \hat{s}\|_\infty \leq \epsilon\}$. Here $\epsilon$ is also referred to as the perturbation budget. In MuJoCo environments, the $\ell_\infty$ norm is applied on normalized state representations.

Table 1: Average rewards ± standard deviation over 50 episodes on three baselines and SA-PPO. We report natural rewards (no attacks) and rewards under five adversarial attacks. In each row we bold the best (lowest) attack reward over all five attacks. The gray rows are the most robust agents.

| Env. | $\epsilon$ | Method | Natural Reward | Attack Reward | | | | | Best Attack |
|---|---|---|---|---|---|---|---|---|---|
| | | | | Critic | Random | MAD | RS | RS+MAD | |
| Hopper | 0.075 | PPO (vanilla) | 3167.6± 541.6 | 1799.0± 935.2 | 2915.2±677.7 | 1505.2± 382.0 | 779.4± 33.2 | **733.8± 44.6** | 733 |
| | | PPO (adv. 50%) | 174± 146 | 69 ±83 | 141± 128 | **42± 46** | 49 ±50 | 44± 43 | 42 |
| | | PPO (adv. 100%) | 6.1± 2.6 | 4.4 ±1.8 | 6.1± 3.2 | 5.8± 2.7 | 3.8 ±0.9 | **3.6 ±0.5** | 3.6 |
| | | SA-PPO (SGLD) | 3523.1±329.0 | 3665.5± 8.2 | 3080.2± 745.4 | 2996.6± 786.4 | **1403.3± 55.0** | 1415.4± 72.0 | 1403.3 |
| | | SA-PPO (Convex) | 3704.1± 2.2 | 3698.4± 4.4 | 3708.7± 23.8 | 3443.1± 466.672 | 1235.8± 50.2 | **1224.2± 47.8** | 1224.2 |
| Walker2d | 0.05 | PPO (vanilla) | 4619.5± 38.2 | 4589.3± 12.4 | 4480.0±465.3 | 4469.1±715.6 | **913.7± 54.3** | 926.8±66.3 | 913.7 |
| | | PPO (adv. 50%) | -11 ± 0.9 | -10.6 ± 0.86 | -10.99 ± 0.95 | -10.78 ± 0.89 | **-11.55 ± 0.79** | -11.37 ± 0.87 | -11.55 |
| | | PPO (adv. 100%) | -113 ± 4.14 | -111.9 ± 4.13 | -111 ± 4.27 | -112 ± 4.08 | -114.4 ± 4.0 | **-114.5 ± 4.09** | -114.5 |
| | | SA-PPO (SGLD) | 4911.8± 188.9 | 5019.0± 65.2 | 4894.8± 139.9 | 4755.7± 413.1 | 2605.6± 1255.7 | **2468.4 ±1205** | 2468.4 |
| | | SA-PPO (Convex) | 4486.6± 60.7 | 4572.0± 52.3 | 4475.0± 48.7 | 4343.4± 329.4 | 2168.2± 665.4 | **2076.1± 666.7** | 2076.1 |
| Humanoid | 0.075 | PPO (vanilla) | 5270.6±1074.3 | 5494.7± 118.7 | 5648.3± 86.8 | 1140.3± 534.8 | 1036.0± 420.2 | **884.1± 356.3** | 884.1 |
| | | PPO (adv. 50%) | 234± 28 | 198 ± 58 | 240 ± 19.4 | 148 ± 73 | **98 ± 69** | 101.5 ± 66.4 | 98 |
| | | PPO (adv. 100%) | 141.4 ± 20.6 | 140.25 ± 16.6 | 142.13 ± 16 | 140.23 ± 34.5 | 113.2 ± 18.5 | **112.6 ± 13.88** | 112.6 |
| | | SA-PPO (SGLD) | 6624.0± 25.5 | 6587.0± 23.1 | 6614.1± 21.4 | 6586.4± 21.3 | 6200.5± 818.1 | **6073.8± 1108.1** | 6073.8 |
| | | SA-PPO (Convex) | 6400.6± 156.8 | 6397.9 ±35.6 | 6207.9± 783.3 | 6379.5± 30.5 | 4707.2± 1359.1 | **4690.3± 1244.89** | 4690.3 |

Figure 4: Box plots of natural and attack rewards for PPO and SA-PPO. Each box is obtained from at least **15 agents** trained with the same parameters as in agents reported in Table 1 The red lines inside the boxes are median rewards, and the upper and lower sides of the boxes show 25% and 75% percentile rewards of 30 agents. The line segments outside of the boxes show min or max rewards.

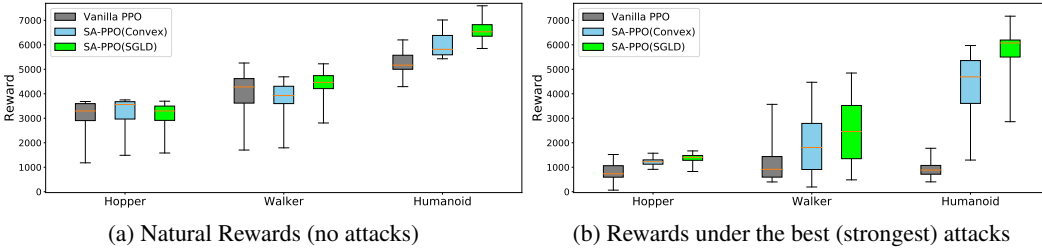

(a) Natural Rewards (no attacks)          (b) Rewards under the best (strongest) attacks

**Evaluation of SA-PPO** We use the PPO implementation from [14], which conducted hyperparameter search and published the optimal hyperparameters for PPO on three Mujoco environments in OpenAI Gym [7]. We use their optimal hyperparameters for PPO, and the same set of hyperparameters for SA-PPO without further tuning. We run Walker2d and Hopper $2 \times 10^6$ steps and Humanoid $1 \times 10^7$ steps to ensure convergence. Our vanilla PPO agents achieve similar or better performance than reported in the literature [14, 25, 22]. Detailed hyperparameters are in Appendix F. SA-PPO has one additional regularization parameter, $\kappa_{\text{PPO}}$, for the regularizer $\mathcal{R}_{\text{PPO}}$, which is chosen in {0.003, 0.01, 0.03, 0.1, 0.3, 1.0}. We solve the SA-PPO objective using both SGLD and convex relaxation methods. We include three baselines: vanilla PPO, and adversarially trained PPO [40, 50] with 50% and 100% training steps under critic attack [50]. The attack is conducted by finding $\hat{s} \in B(s)$ minimizing $V(\hat{s})$ instead of $Q(s, \pi(\hat{s}))$, as PPO does not learn a $Q$ function during learning. We evaluate agents using 5 attacks, including our strong RS and MAD attacks, detailed in Appendix D.

In Table 1, naive adversarial training deteriorates performance and does not reliably improve robustness in all three environments. Our RS attack and MAD attacks are very effective in all environments and achieve significantly lower rewards than critic and random attacks; this shows the importance of evaluation using strong attacks. SA-PPO, solved either by SGLD or the convex relaxation objective, *significantly improves robustness* against strong attacks. Additionally, SA-PPO achieves natural performance (without attacks) similar to that of vanilla PPO in Walker2d and Hopper, and *significantly improves the reward in Humanoid environment*. Humanoid has a high state-space dimension (376) and is usually hard to train [22], and our results suggest that a robust objective can be helpful even in a non-adversarial setting. Because PPO training can have large performance variance across multiple runs, to show that our SA-PPO can consistently obtain a robust agent, we repeatedly train each environment using SA-PPO and vanilla PPO at least **15 times** and attack all agents obtained. In Figures 4a and 4b we show the box plot of the natural and best attack reward for these PPO and SA-PPO agents. We can see that the best attack rewards of most SA-PPO agents are consistently better than PPO agents (in terms of median, 25% and 75% percentile rewards over multiple repetitions).

**Evaluation of SA-DDPG** We use a high quality DDPG implementation [61] as our baseline, achieving similar or better performance on five Mujoco environments as in the literature [35, 17]. For SA-DDPG, we use the same set of hyperparameters as in DDPG [61] (detailed in Appendix G), except for the additional regularization term $\kappa_{\text{DDPG}}$ for $\mathcal{R}_{\text{DDPG}}$ which is searched in {0.1, 0.3, 1.0, 3.0} for

Table 2: Average rewards $\pm$ standard deviation over 50 episodes on DDPG, adversarial training [50] (50% and 100% steps) and SA-DDPG. Each number represents an agent with *median* reward under the best attack over 11 training runs with identical hyperparameters. Due to large variance in RL, it important to report median metrics. **Bold** numbers indicate the most robust agents. Full results of all five attacks are in Table 6 and results over multiple training runs are in Table 12 (Appendix I).

| Environment | | Ant | Hopper | Inverted Pendulum | Reacher | Walker2d |
|---|---|---|---|---|---|---|
| $\ell_\infty$ norm perturbation budget $\epsilon$ | | 0.2 | 0.075 | 0.3 | 1.5 | 0.05 |
| DDPG (vanilla) | Natural Reward | $1487 \pm 850$ | $3302 \pm 762$ | $1000 \pm 0$ | $-4.37 \pm 1.54$ | $1870 \pm 1418$ |
| | Attack Reward (best) | $142 \pm 180$ | $606 \pm 124$ | $92 \pm 1$ | $-27.87 \pm 4.38$ | $790 \pm 985$ |
| DDPG (adv. 50%) | Natural Reward | $1487 \pm 850$ | $3302 \pm 762$ | $1000 \pm 0$ | $-4.37 \pm 1.54$ | $1870 \pm 1418$ |
| | Attack Reward (best) | $31 \pm 179$ | $41 \pm 105$ | $39 \pm 0$ | $-25.81 \pm 6.53$ | $837 \pm 722$ |
| DDPG (adv. 100%) | Natural Reward | $1082 \pm 574$ | $973 \pm 0$ | $1000 \pm 0$ | $-5.71 \pm 1.80$ | $462 \pm 569$ |
| | Attack Reward (best) | $-52 \pm 231$ | $24 \pm 15$ | $82 \pm 0$ | $-27.44 \pm 4.05$ | $302 \pm 260$ |
| SA-DDPG (SGLD) | Natural Reward | $2186 \pm 534$ | $3068 \pm 223$ | $1000 \pm 0$ | $-5.38 \pm 1.74$ | $3318 \pm 680$ |
| | Attack Reward (best) | $\mathbf{2007 \pm 686}$ | $\mathbf{1609 \pm 676}$ | $423 \pm 281$ | $\mathbf{-12.10 \pm 4.58}$ | $1210 \pm 979$ |
| SA-DDPG (convex relax) | Natural Reward | $2254 \pm 430$ | $3128 \pm 453$ | $1000 \pm 0$ | $-5.24 \pm 2.06$ | $4540 \pm 1562$ |
| | Attack Reward (best) | $1820 \pm 635$ | $1202 \pm 402$ | $\mathbf{1000 \pm 0}$ | $-12.44 \pm 3.77$ | $\mathbf{1986 \pm 1993}$ |

Table 3: Average rewards $\pm$ std and action certification rate over 50 episodes on three baselines and SA-DQN. We report natural rewards (no attacks) and PGD attack rewards (under 10-step or 50-step PGD). Action Cert. Rate is the proportion of the actions during rollout that are guaranteed unchanged by any attacks within the given $\epsilon$. Training time is reported in Section H.

| Environment | | Pong | Freeway | BankHeist | RoadRunner |
|---|---|---|---|---|---|
| $\ell_\infty$ norm perturbation budget $\epsilon$ | | 1/255 | | | |
| DQN (vanilla) | Natural Reward | $21.0 \pm 0.0$ | $34.0 \pm 0.2$ | $1308.4 \pm 24.1$ | $45534.0 \pm 7066.0$ |
| | PGD Attack Reward (10 steps) | $-21.0 \pm 0.0$ | $0.0 \pm 0.0$ | $56.4 \pm 21.2$ | $0.0 \pm 0.0$ |
| | Action Cert. Rate | 0.0 | 0.0 | 0.0 | 0.0 |
| DQN Adv. Training (attack 50% frames) Behzadan & Munir [5] | Natural Reward | $10.1 \pm 6.6$ | $25.4 \pm 0.8$ | $1126.0 \pm 70.9$ | $22944.0 \pm 6532.5$ |
| | PGD Attack Reward (10 steps) | $-21.0 \pm 0.0$ | $0.0 \pm 0.0$ | $9.4 \pm 13.6$ | $14.0 \pm 34.7$ |
| | Action Cert. Rate | 0.0 | 0.0 | 0.0 | 0.0 |
| Imitation learning Fischer et al. [15] | Natural Reward | 19.73 | 32.93 | 238.66 | 12106.67 |
| | PGD Attack Reward (4 steps) | 18.13 | **32.53** | 190.67 | 5753.33 |
| SA-DQN (PGD) | Natural Reward | $21.0 \pm 0.0$ | $33.9 \pm 0.4$ | $1245.2 \pm 14.5$ | $34032.0 \pm 3845.0$ |
| | PGD Attack Reward (10 steps) | $21.0 \pm 0.0$ | $23.7 \pm 2.3$ | $1006.0 \pm 226.4$ | $20402.0 \pm 7551.1$ |
| | Action Cert. Rate | 0.0 | 0.0 | 0.0 | 0.0 |
| SA-DQN (convex) | Natural Reward | $21.0 \pm 0.0$ | $30.0 \pm 0.0$ | $1235.4 \pm 9.8$ | $44638.0 \pm 7367.0$ |
| | PGD Attack Reward (10 steps) | $\mathbf{21.0 \pm 0.0}$ | $30.0 \pm 0.0$ | $\mathbf{1232.4 \pm 16.2}$ | $\mathbf{44732.0 \pm 8059.5}$ |
| | PGD Attack Reward (50 steps) | $\mathbf{21.0 \pm 0.0}$ | $30.0 \pm 0.0$ | $\mathbf{1234.6 \pm 16.6}$ | $\mathbf{44678.0 \pm 6954.0}$ |
| | Action Cert. Rate | 1.000 | 1.000 | 0.984 | 0.475 |

InvertedPendulum and Reacher due to their low dimensionality and $\{30, 100, 300, 1000\}$ for other environments. We include vanilla DDPG, adversarially trained DDPG [50] (attacking 50% or 100% steps) as baselines. We use the same set of 5 attacks as in 1. In Table 2, we observe that naive adversarial training is not very effective in many environments. SA-DDPG (solved by SGLD or convex relaxations) significantly improves robustness under strong attacks in all 5 environments. Similar to the observations on SA-PPO, SA-DDPG can improve natural agent performance in environments (Ant and Walker2d) with relatively high dimensional state space (111 and 17).

**Evaluation of SA-DQN** We implement Double DQN [72] and Prioritized Experience Replay [58] on four Atari games. We train Atari agents for 6 million frames for both vanilla DQN and SA-DQN. Detailed parameters and training procedures are in Appendix H. We normalize the pixel values to $[0, 1]$ and we add $\ell_\infty$ adversarial noise with norm $\epsilon = 1/255$. We include vanilla DQNs and adversarially trained DQNs with 50% of frames under attack [5] during training time as baselines, and we report results of robust imitation learning [15]. We evaluate all environments under 10-step untargeted PGD attacks, except that results from [15] were evaluated using a weaker 4-step PGD attack. For the most robust Atari agents (SA-DQN convex), we additionally attack them using 50-step PGD attacks, and find that the rewards do not further reduce. In Table 3, we see that our SA-DQN achieves much higher rewards under attacks in most environments, and naive adversarial training is mostly ineffective under strong attacks. We obtain better rewards than [15] in most environments, as we learn the agents directly rather than using two-step imitation learning.

**Robustness certificates.** When our robust policy regularizer is trained using convex relaxations, we can obtain certain robustness certificates under observation perturbations. For a simple environment like Pong, we can guarantee actions do not change for all frames during rollouts, thus guarantee the cumulative rewards under perturbation. For SA-DDPG, the *upper bounds* on the maximal $\ell_2$ difference in action changes is a few times smaller than baselines on all 5 environments (see Appendix I). Unfortunately, for most RL tasks, due to the complexity of environment dynamics and reward process, it is impossible to obtain a "certified reward" as the certified test error in supervised learning settings [79, 88]. We leave further discussions on these challenges in Appendix E.

## Broader Impact

Reinforcement learning is a central part of modern artificial intelligence and is still under heavy development in recent years. Unlike supervised learning which has been widely deployed in many commercial and industrial applications, reinforcement learning has not been widely accepted and deployed in real-world settings. Thus, the study of reinforcement learning robustness under the adversarial attacks settings receives less attentions than the supervised learning counterparts.

However, with the recent success of reinforcement learning on many complex games such as Go [65], StartCraft [73] and Dota 2 [6], we will not be surprised if we will see reinforcement learning (especially, deep reinforcement learning) being used in everyday decision making tasks in near future. The potential social impacts of applying reinforcement learning agents thus must be investigated before its wide deployment. One important aspect is the trustworthiness of an agent, where robustness plays a crucial rule. The robustness considered in our paper is important for many realistic settings such as sensor noise, measurement errors, and man-in-the-middle (MITM) attacks for a DRL system. if the robustness of reinforcement learning can be established, it has the great potential to be applied into many mission-critical tasks such as autonomous driving [60, 56, 85] to achieve superhuman performance.

On the other hand, one obstacle for applying reinforcement learning to real situations (beyond games like Go and StarCraft) is the "reality gap": a well trained reinforcement learning agent in a simulation environment can easily fail in real-world experiments. One reason for this failure is the potential sensing errors in real-world settings; this was discussed as early as in Brooks [8] in 1992 and still remains an open challenge now. Although our experiments were done in simulated environments, we believe that a smoothness regularizer like the one proposed in our paper can also benefit agents tested in real-world settings, such as robot hand manipulation [2].

## Acknowledgments and Disclosure of Funding

We acknowledge the support by NSF IIS-1901527, IIS-2008173, ARL-0011469453, and scholarship by IBM. The authors thank Ge Yang and Xiaocheng Tang for helpful discussions.

## Footnotes

[1]Our analysis also holds for a stochastic adversary. The optimal adversary is deterministic (see Lemma 1).

[2]Code and pretrained agents available at https://github.com/chenhongge/StateAdvDRL

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
