[Supplementary Material]

- Readers who are interested in SA-MDP can find an example of SA-MDP in Section A and complete proofs in Section B.

- Readers who are interested in adversarial attacks can find more details about our new attacks and existing attacks in Section D. Especially, we discussed how a robust critic can help in attacking RL, and show experiments on the improvements gained by the robustness objective during attack.

- Readers who want to know more details of optimization techniques to solve our state-adversarial robust regularizers can refer to Section C, including more background on convex relaxations of neural networks in Section C.1.

- We provide detailed algorithm and hyperparameters for SA-PPO in Section F. We provide details for SA-DDPG in Section G. We provide details for SA-DQN in Section H.

- We provide more empirical results in Section I. To demonstrate the convergence of our algorithm, we repeat each experiment at least 15 times and plot the convergence of rewards during multiple runs. We found that for some environments (like Humanoid) we can consistently improve baseline performance. We also evaluate some settings under multiple perturbation strength $\epsilon$.

## A  An example of SA-MDP

We first show a simple environment and solve it under different settings of MDP and SA-MDP. The environment has three states $\mathcal{S} = \{S_1, S_2, S_3\}$ and 2 actions $\mathcal{A} = \{A_1, A_2\}$. The transition probabilities and rewards are defined as below (unmentioned probabilities and rewards are 0):

$$\Pr(s' = S_1 | s = S_1, a = A_1) = 1.0$$
$$\Pr(s' = S_2 | s = S_1, a = A_2) = 1.0$$
$$\Pr(s' = S_2 | s = S_2, a = A_2) = 1.0$$
$$\Pr(s' = S_3 | s = S_2, a = A_1) = 1.0$$
$$\Pr(s' = S_1 | s = S_3, a = A_2) = 1.0$$
$$\Pr(s' = S_2 | s = S_3, a = A_1) = 1.0$$
$$R(s = S_1, a = A_2, s' = S_2) = 1.0$$
$$R(s = S_2, a = A_1, s' = S_2) = 1.0$$
$$R(s = S_3, a = A_1, s' = S_3) = 1.0$$

The environment is illustrated in Figure 5. For the power of adversary, we allow $\nu$ to perturb one

Figure 5: A simple 3-state toy environment.

state to any other two neighbouring states:

$$B_\nu(S_1) = B_\nu(S_2) = B_\nu(S_3) = \{S_1, S_2, S_3\}$$

Now we evaluate various policies for MDP and SA-MDP for this environment. We use $\gamma = 0.99$ as the discount factor. A stationary and Markovian policy in this environment can be described by 3 parameters $p_{11}, p_{21}, p_{31}$ where $p_{ij} \in [0, 1]$ denotes the probability $\Pr(a = A_j | s = S_i)$. We denote the value function as $V$ for MDP and $\tilde{V}$ for SA-MDP.

- **Optimal Policy for MDP.** For a regular MDP, the optimal solution is $p_{11} = 0$, $p_{21} = 1$, $p_{31} = 1$. We take $A_2$ to receive reward and leave $S_1$, and then keep doing $A_1$ in $S_2$ and $S_3$. The values for each state are $V(S_1) = V(S_2) = V(S_3) = \frac{1}{1-\gamma} = 100$, which is optimal. However, this policy obtains $\tilde{V}(S_1) = \tilde{V}(S_2) = \tilde{V}(S_3) = 0$ for SA-MDP, because we can set $\nu(S_1) = S_2$, $\nu(S_2) = S_1$, $\nu(S_3) = S_1$ and consequentially we always take the wrong action receiving 0 reward.

- **A Stochastic Policy for MDP and SA-MDP.** We consider a stochastic policy where $p_{11} = p_{21} = p_{31} = 0.5$. Under this policy, we randomly stay or move in each state, and has a 50% probability of receiving a reward. The adversary $\nu$ has no power because $\pi$ is the same for all states. In this situation, $V(S_1) = \tilde{V}(S_1) = V(S_2) = \tilde{V}(S_2) = V(S_3) = \tilde{V}(S_3) = \frac{0.5}{1-0.99} = 50$ for both MDP and SA-MDP. This can also be seen as an extreme case of Theorem 5, where the policy does not change under adversary in all states, so there is no performance loss in SA-MDP.

- **Deterministic Policies for SA-MDP.** Now we consider all $2^3 = 8$ possible deterministic policies for SA-MDP. Note that if for any state $S_i$ we have $p_{i1} = 0$ and another state $S_j$ we have $p_{j1} = 1$, we always have $\tilde{V}(S_1) = \tilde{V}(S_2) = \tilde{V}(S_3) = 0$. This is because we can set $\nu(S_1) = S_j$, $\nu(S_2) = S_i$ and $\nu(S_3) = S_i$ and always receive a 0 reward. Thus the only two possible other policies are $p_{11} = p_{21} = p_{31} = 0$ and $p_{11} = p_{21} = p_{31} = 1$, respectively. For $p_{11} = p_{21} = p_{31} = 1$ we have $\tilde{V}(S_1) = 0$, $\tilde{V}(S_2) = \tilde{V}(S_3) = 100$ as we always take $A_1$ and never transit to other states; for $p_{11} = p_{21} = p_{31} = 0$, we circulate through all three states and only receive a reward when we leave $A_1$. We have $\tilde{V}(S_1) = \frac{1}{1-\gamma^3} \approx 33.67$, $\tilde{V}(S_2) = \frac{\gamma^2}{1-\gamma^3} \approx 33.00$ and $\tilde{V}(S_3) = \frac{\gamma}{1-\gamma^3} \approx 33.33$.

Figure 6, 7, 8 give the graphs of $\tilde{V}(S_1)$, $\tilde{V}(S_2)$ and $\tilde{V}(S_3)$ under three different settings of $p_{11}$. The figures are generated using Algorithm 1.

Figure 6: Value functions for SA-MDP when $p_{11} = 0$, with $p_{21} \in [0,1]$, $p_{31} \in [0,1]$

Figure 7: Value functions for SA-MDP when $p_{11} = 0.5$, with different $p_{21} \in [0,1]$, $p_{31} \in [0,1]$

Figure 8: Value functions for SA-MDP when $p_{11} = 1.0$, with different $p_{21} \in [0, 1]$, $p_{31} \in [0, 1]$

# B  Proofs for State-Adversarial Markov Decision Process

**Theorem 1** (Bellman equations for fixed $\pi$ and $\nu$). *Given $\pi : \mathcal{S} \to \mathcal{P}(\mathcal{A})$ and $\nu : \mathcal{S} \to \mathcal{S}$, we have*

$$\tilde{V}_{\pi \circ \nu}(s) = \sum_{a \in \mathcal{A}} \pi(a|\nu(s)) \sum_{s' \in \mathcal{S}} p(s'|s, a) \left[ R(s, a, s') + \gamma \tilde{V}_{\pi \circ \nu}(s') \right]$$

$$\tilde{Q}_{\pi \circ \nu}(s, a) = \sum_{s' \in \mathcal{S}} p(s'|s, a) \left[ R(s, a, s') + \gamma \sum_{a' \in \mathcal{A}} \pi(a'|\nu(s')) \tilde{Q}_{\pi \circ \nu}(s', a') \right].$$

*Proof.* Based on the definition of $\tilde{V}_{\pi \circ \nu}(s)$:

$$
\begin{aligned}
\tilde{V}_{\pi \circ \nu}(s) &= \mathbb{E}_{\pi \circ \nu} \left[ \sum_{k=0}^{\infty} \gamma^k r_{t+k+1} | s_t = s \right] \\
&= \mathbb{E}_{\pi \circ \nu} \left[ r_{t+1} + \gamma \sum_{k=0}^{\infty} \gamma^k r_{t+k+2} | s_t = s \right] \\
&= \sum_{a \in \mathcal{A}} \pi(a|\nu(s)) \sum_{s' \in \mathcal{S}} p(s'|s, a) \left[ r_{t+1} + \gamma \mathbb{E}_{\pi \circ \nu} \left[ \sum_{k=0}^{\infty} \gamma^k r_{t+k+2} | s_{t+1} = s' \right] \right] \\
&= \sum_{a \in \mathcal{A}} \pi(a|\nu(s)) \sum_{s' \in \mathcal{S}} p(s'|s, a) \left[ R(s, a, s') + \gamma \tilde{V}_{\pi \circ \nu}(s') \right]
\end{aligned}
\tag{10}
$$

The recursion for $\tilde{Q}_{\pi \circ \nu}(s, a)$ can be derived similarly. Additionally, we note the following useful relationship between $\tilde{V}_{\pi \circ \nu}(s)$ and $\tilde{Q}_{\pi \circ \nu}(s, a)$:

$$\tilde{V}_{\pi \circ \nu}(s) = \sum_{a \in \mathcal{A}} \pi(a|\nu(s)) \tilde{Q}_{\pi \circ \nu}(s, a) \tag{11}$$

$\square$

Before starting to prove Theorem 2, first we show that finding the optimal adversary $\nu^*$ given a fixed $\pi$ for a SA-MDP can be cast into the problem of finding an optimal policy in a regular MDP.

**Lemma 1** (Equivalence of finding optimal adversary in SA-MDP and finding optimal policy in MDP). *Given an SA-MDP $M = (\mathcal{S}, \mathcal{A}, B, R, p, \gamma)$ and a fixed policy $\pi$, there exists a MDP $\hat{M} = (\mathcal{S}, \hat{\mathcal{A}}, \hat{R}, \hat{p}, \gamma)$ such that the optimal policy of $\hat{M}$ is the optimal adversary $\nu$ for SA-MDP given the fixed $\pi$.*

*Proof.* For an SA-MDP $M = (\mathcal{S}, \mathcal{A}, B, R, p, \gamma)$ and a fixed policy $\pi$, we define a regular MDP $\hat{M} = (\mathcal{S}, \hat{\mathcal{A}}, \hat{R}, \hat{p}, \gamma)$ such that $\hat{\mathcal{A}} = \mathcal{S}$, and $\nu$ is the policy for $\hat{M}$. At each state $s$, our policy $\nu$ gives a probability distribution $\nu(\cdot|s) \in \mathcal{P}(\hat{\mathcal{A}}) = \mathcal{P}(\mathcal{S})$ indicating that we perturb a state $s$ to $\hat{s}$ with probability $\nu(\hat{s}|s)$ in the SA-MDP $M$.

For $\hat{M}$, the reward function is defined as:

$$
\hat{R}(s, \hat{a}, s') = \begin{cases} -\dfrac{\sum_{a \in \mathcal{A}} \pi(a|\hat{a}) p(s'|s, a) R(s, a, s')}{\sum_{a \in \mathcal{A}} \pi(a|\hat{a}) p(s'|a, s)} & \text{for } s, s' \in \mathcal{S} \text{ and } \hat{a} \in B(s) \subset \hat{\mathcal{A}} = \mathcal{S}, \\ C & \text{for } s, s' \in \mathcal{S} \text{ and } \hat{a} \notin B(s). \end{cases}
\tag{12}
$$

The transition probability $\hat{p}$ is defined as

$$\hat{p}(s'|s, \hat{a}) = \sum_{a \in \mathcal{A}} \pi(a|\hat{a})p(s'|s, a) \quad \text{for } s, s' \in \mathcal{S} \text{ and } \hat{a} \in \hat{\mathcal{A}} = \mathcal{S}.$$

The above reward function definition is based on the following conditional probability which marginalizes $\pi$:

$$
\begin{aligned}
p(r|s, \hat{a}, s') &= \frac{p(r, s'|s, \hat{a})}{p(s'|s, \hat{a})} \\
&= \frac{\sum_a p(r, s'|a, s, \hat{a})\pi(a|s, \hat{a})}{\sum_a p(s'|a, s, \hat{a})\pi(a|s, \hat{a})} \\
&= \frac{\sum_a p(r, s'|a, s)\pi(a|\hat{a})}{\sum_a p(s'|a, s)\pi(a|\hat{a})} \\
&= \frac{\sum_a p(r|s', a, s)p(s'|a, s)\pi(a|\hat{a})}{\sum_a p(s'|a, s)\pi(a|\hat{a})}
\end{aligned}
$$

Considering that $p(r = R(S, A, S')|s' = S', a = A, s = S) = 1.0$ and 0 otherwise, and taking an expectation over $r$ yields the first case in (12).

The reward for adversary's actions outside $B(s)$ is a constant $C$ such that

$$C < \min\left\{-\overline{R}, \quad \frac{\gamma}{(1-\gamma)}\underline{R} - \frac{1}{(1-\gamma)}\overline{R}\right\},$$

where $\underline{R} := \min_{s, \hat{a}, s'} R(s, \hat{a}, s')$ and $\overline{R} := \max_{s, \hat{a}, s'} R(s, \hat{a}, s')$. We have for $\forall \hat{a}$,

$$C < \hat{R}(s, \hat{a}, s') \leq -\underline{R},$$

and for $\forall \hat{a} \in B(s)$,

$$-\overline{R} \leq \hat{R}(s, \hat{a}, s') \leq -\underline{R}.$$

According to results on MDP [53, 68], we know that the $\hat{M}$ has an optimal policy $\nu^*$, which satisfies $\hat{V}_{\pi \circ \nu^*}(s) \geq \hat{V}_{\pi \circ \nu}(s)$ for $\forall s$, $\forall \nu$. We also know that this $\nu^*$ is deterministic and assigns a unit mass probability for the optimal action for each $s$.

We define $\mathfrak{N} := \{\nu : \forall s, \exists \hat{a} \in B(s), \nu(\hat{a}|s) = 1\}$ and claim that $\nu^* \in \mathfrak{N}$. If this is not true for a state $s^0$, we have

$$
\begin{aligned}
\hat{V}_{\pi \circ \nu^*}(s^0) &= \mathbb{E}_{\hat{p}, \nu^*}\left[\sum_{k=0}^{\infty} \gamma^k \hat{r}_{t+k+1}|s_t = s^0\right] \\
&= C + \mathbb{E}_{\hat{p}, \nu^*}\left[\sum_{k=1}^{\infty} \gamma^k \hat{r}_{t+k+1}|s_t = s^0\right] \\
&\leq C - \frac{\gamma}{1-\gamma}\underline{R} \\
&< -\frac{1}{1-\gamma}\overline{R} \\
&\leq \mathbb{E}_{\hat{p}, \nu'}\left[\sum_{k=0}^{\infty} \gamma^k \hat{r}_{t+k+1}|s_t = s^0\right] = \hat{V}_{\pi \circ \nu'}(s^0),
\end{aligned}
$$

where the second equality holds because $\nu^*$ is deterministic, and the last inequality holds for any $\nu' \in \mathfrak{N}$. This contradicts the assumption that $\nu^*$ is optimal. So from now on in this proof we only study policies in $\mathfrak{N}$.

For any policy $\nu \in \mathfrak{N}$ :

$$\hat{V}_{\pi \circ \nu}(s) = \mathbb{E}_{\hat{p}, \nu}\left[\sum_{k=0}^{\infty} \gamma^k \hat{r}_{t+k+1} | s_t = s\right]$$

$$= \mathbb{E}_{\hat{p}, \nu}\left[\hat{r}_{t+1} + \gamma \sum_{k=0}^{\infty} \gamma^k \hat{r}_{t+k+2} | s_t = s\right]$$

$$= \sum_{\hat{a} \in \mathcal{S}} \nu(\hat{a}|s) \sum_{s' \in \mathcal{S}} \hat{p}(s'|s, \hat{a})\left[\hat{R}(s, \hat{a}, s') + \gamma \mathbb{E}_{\hat{p}, \nu}\left[\sum_{k=0}^{\infty} \gamma^k \hat{r}_{t+k+2} | s_{t+1} = s'\right]\right]$$

$$= \sum_{\hat{a} \in \mathcal{S}} \nu(\hat{a}|s) \sum_{s' \in \mathcal{S}} \hat{p}(s'|s, \hat{a})\left[\hat{R}(s, \hat{a}, s') + \gamma \hat{V}_{\pi \circ \nu}(s')\right] \tag{13}$$

Note that all policies in $\mathfrak{N}$ are deterministic and this class of policies consists $\nu^*$. Also, $\mathfrak{N}$ is consistent with the class of policies studied in Theorem 1. We denote the deterministic action $\hat{a}$ chosen by a $\nu \in \mathfrak{N}$ at $s$ as $\nu(s)$. Then for $\forall \nu \in \mathfrak{N}$, we have

$$\hat{V}_{\pi \circ \nu}(s) = \sum_{s' \in \mathcal{S}} \hat{p}(s'|s, \nu(s))\left[\hat{R}(s, \hat{a}, s') + \gamma \hat{V}_{\pi \circ \nu}(s')\right]$$

$$= \sum_{s' \in \mathcal{S}} \sum_{a \in \mathcal{A}} \pi(a|\hat{a}) p(s'|s, a)\left[-\frac{\sum_{a \in \mathcal{A}} \pi(a|\hat{a}) p(s'|s, a) R(s, a, s')}{\sum_{a \in \mathcal{A}} \pi(a|\hat{a}) p(s'|a, s)} + \gamma \hat{V}_{\pi \circ \nu}(s')\right]$$

$$= \sum_{a \in \mathcal{A}} \pi(a|\nu(s)) \sum_{s' \in \mathcal{S}} p(s'|s, a)\left[-R(s, a, s') + \gamma \hat{V}_{\pi \circ \nu}(s')\right], \tag{14}$$

or

$$-\hat{V}_{\pi \circ \nu}(s) = \sum_{a \in \mathcal{A}} \pi(a|\nu(s)) \sum_{s' \in \mathcal{S}} p(s'|s, a)\left[R(s, a, s') + \gamma(-\hat{V}_{\pi \circ \nu}(s'))\right]. \tag{15}$$

Comparing (15) and (10), we know that $-\hat{V}_{\pi \circ \nu} = \tilde{V}_{\pi \circ \nu}$ for any $\nu \in \mathfrak{N}$. The optimal value function $\hat{V}_{\pi \circ \nu^*}$ satisfies:

$$\hat{V}_{\pi \circ \nu^*}(s) = \max_{\hat{a} \in B(s)} \sum_{s' \in \mathcal{S}} \hat{p}(s'|s, \hat{a})\left[\hat{R}(s, \hat{a}, s') + \gamma \hat{V}_{\pi \circ \nu}(s')\right]$$

$$= \max_{s_\nu \in B(s)} \sum_{a \in \mathcal{A}} \pi(a|s_\nu) \sum_{s' \in \mathcal{S}} p(s'|s, a)\left[-R(s, a, s') + \gamma \hat{V}_{\pi \circ \nu^*}(s')\right], \tag{16}$$

where we denote the action $\hat{a}$ taken at $s$ as $s_\nu$. So for $\nu^*$, since $-\hat{V}_{\pi \circ \nu^*} = \tilde{V}_{\pi \circ \nu^*}$, we have

$$\tilde{V}_{\pi \circ \nu^*}(s) = \min_{\hat{a} \in B(s)} \sum_{a \in \mathcal{A}} \pi(a|\hat{a}) \sum_{s' \in \mathcal{S}} p(s'|s, a)\left[R(s, a, s') + \gamma \tilde{V}_{\pi \circ \nu^*}(s')\right], \tag{17}$$

and $\tilde{V}_{\pi \circ \nu^*}(s) \leq \tilde{V}_{\pi \circ \nu}(s)$ for $\forall s$, $\forall \nu \in \mathfrak{N}$. Hence $\nu^*$ is also the optimal $\nu$ for $\tilde{V}_{\pi \circ \nu}$. $\qquad \square$

Lemma 1 gives many good properties for the optimal adversary. First, an optimal adversary always exists. Second, we do not need to consider stochastic adversaries as there always exists an optimal deterministic adversary. Additionally, showing Bellman contraction for finding the optimal adversary can be done similarly as in obtaining the optimal policy in a regular MDP, as shown in the proof of Theorem 2.

**Theorem 2** (Bellman contraction for optimal adversary). *Define Bellman operator* $\mathscr{L} : \mathbb{R}^{|\mathcal{S}|} \to \mathbb{R}^{|\mathcal{S}|}$,

$$(\mathscr{L}\tilde{V})(s) = \min_{s_\nu \in B(s)} \sum_{a \in \mathcal{A}} \pi(a|s_\nu) \sum_{s' \in \mathcal{S}} p(s'|s, a)\left[R(s, a, s') + \gamma \tilde{V}(s')\right]. \tag{18}$$

*The Bellman equation for optimal adversary* $\nu^*$ *can then be written as:* $\tilde{V}_{\pi \circ \nu^*} = \mathscr{L}\tilde{V}_{\pi \circ \nu^*}$. *Additionally,* $\mathscr{L}$ *is a contraction that converges to* $\tilde{V}_{\pi \circ \nu^*}$.

*Proof.* Based on Lemma 1, this proof is technically similar to the proof of "optimal Bellman equation" in regular MDPs, where $\max$ over $\pi$ is replaced by $\min$ over $\nu$. By the definition of $\tilde{V}_{\pi\circ\nu^*}(s)$,

$$\tilde{V}_{\pi\circ\nu^*}(s) = \min_\nu \tilde{V}_{\pi\circ\nu}(s)$$

$$= \min_\nu \mathbb{E}_{\pi\circ\nu}\left[\sum_{k=0}^\infty \gamma^k r_{t+k+1}|s_t = s\right]$$

$$= \min_\nu \mathbb{E}_{\pi\circ\nu}\left[r_{t+1} + \gamma\sum_{k=0}^\infty \gamma^k r_{t+k+2}|s_t = s\right]$$

$$= \min_\nu \sum_{a\in\mathcal{A}} \pi(a|\nu(s)) \sum_{s'\in\mathcal{S}} p(s'|s,a)\left[r_{t+1} + \gamma\mathbb{E}_{\pi\circ\nu}\left[\sum_{k=0}^\infty \gamma^k r_{t+k+2}|s_{t+1} = s'\right]\right]$$

$$= \min_{s_\nu\in B_\nu(s)} \sum_{a\in\mathcal{A}} \pi(a|s_\nu) \sum_{s'\in\mathcal{S}} p(s'|s,a)\left[r_{t+1} + \gamma\min_\nu\mathbb{E}_{\pi\circ\nu}\left[\sum_{k=0}^\infty \gamma^k r_{t+k+2}|s_{t+1} = s'\right]\right]$$

$$= \min_{s_\nu\in B_\nu(s)} \sum_{a\in\mathcal{A}} \pi(a|s_\nu) \sum_{s'\in\mathcal{S}} p(s'|s,a)\left[r_{t+1} + \gamma\tilde{V}_{\pi\circ\nu^*}(s')\right]$$

This is the Bellman equation for the optimal adversary $\nu^*$; $\nu^*$ is a fixed point of the Bellman operator $\mathscr{L}$.

Now we show the Bellman operator is a contraction. We have, if $\mathscr{L}\tilde{V}_{\pi\circ\nu_1}(s) \geq \mathscr{L}\tilde{V}_{\pi\circ\nu_2}(s)$,

$$\mathscr{L}\tilde{V}_{\pi\circ\nu_1}(s) - \mathscr{L}\tilde{V}_{\pi\circ\nu_2}(s)$$

$$\leq \max_{s_\nu\in B_\nu(s)}\left\{\sum_{a\in\mathcal{A}} \pi(a|s_\nu) \sum_{s'\in\mathcal{S}} p(s'|s,a)\left[R(s,a,s') + \gamma\tilde{V}_{\pi\circ\nu_1}(s')\right]\right.$$

$$\left. - \sum_{a\in\mathcal{A}} \pi(a|s_\nu) \sum_{s'\in\mathcal{S}} p(s'|s,a)\left[R(s,a,s') + \gamma\tilde{V}_{\pi\circ\nu_2}(s')\right]\right\}$$

$$= \gamma\max_{s_\nu\in B_\nu(s)} \sum_{a\in\mathcal{A}} \pi(a|s_\nu) \sum_{s'\in\mathcal{S}} p(s'|s,a)[\tilde{V}_{\pi\circ\nu_1}(s') - \tilde{V}_{\pi\circ\nu_2}(s')]$$

$$\leq \gamma\max_{s_\nu\in B_\nu(s)} \sum_{a\in\mathcal{A}} \pi(a|s_\nu) \sum_{s'\in\mathcal{S}} p(s'|s,a)\|\tilde{V}_{\pi\circ\nu_1} - \tilde{V}_{\pi\circ\nu_2}\|_\infty$$

$$= \gamma\|\tilde{V}_{\pi\circ\nu_1} - \tilde{V}_{\pi\circ\nu_2}\|_\infty$$

The first inequality comes from the fact that

$$\min_{x_1} f(x_1) - \min_{x_2} g(x_2) \leq f(x_2^*) - g(x_2^*) \leq \max_x(f(x) - g(x)),$$

where $x_2^* = \arg\min_{x_2} g(x_2)$. Similarly, we can prove $\mathscr{L}\tilde{V}_{\pi\circ\nu_2}(s) - \mathscr{L}\tilde{V}_{\pi\circ\nu_1}(s) \leq \|\tilde{V}_{\pi\circ\nu_1} - \tilde{V}_{\pi\circ\nu_2}\|_\infty$ if $\mathscr{L}\tilde{V}_{\pi\circ\nu_2}(s) > \mathscr{L}\tilde{V}_{\pi\circ\nu_1}(s)$. Hence

$$\|\mathscr{L}\tilde{V}_{\pi\circ\nu_1}(s) - \mathscr{L}\tilde{V}_{\pi\circ\nu_2}(s)\|_\infty = \max_s |\mathscr{L}\tilde{V}_{\pi\circ\nu_1}(s) - \mathscr{L}\tilde{V}_{\pi\circ\nu_2}(s)| \leq \gamma\|\tilde{V}_{\pi\circ\nu_1} - \tilde{V}_{\pi\circ\nu_2}\|_\infty.$$

Then according to the Banach fixed-point theorem, since $0 < \gamma < 1$, $\tilde{V}_{\pi\circ\nu}$ converges to a unique fixed point, and this fixed point is $\tilde{V}_{\pi\circ\nu^*}$.

$\square$

A direct consequence of Theorem 2 is the policy evaluation algorithm (Algorithm 1) for SA-MDP, which obtains the values for each state under *optimal* adversary for a fixed policy $\pi$. For both Lemma 1 and Theorem 2, we only consider a fixed policy $\pi$, and in this setting finding an optimal adversary is not difficult. However, finding an optimal $\pi$ under the optimal adversary is more challenging, as we can see in Section A, given the white-box attack setting where the adversary knows $\pi$ and can choose optimal perturbations accordingly, an optimal policy for MDP can only receive zero rewards under optimal adversary. We now show two intriguing properties for optimal policies in SA-MDP:

**Algorithm 1** Policy Evaluation for an SA-MDP $(\mathcal{S}, \mathcal{A}, B, R, p, \gamma)$

---

**Input:** Policy $\pi$, convergence threshold $\varepsilon$
**Output:** Values for policy $\pi$, detnoted as $\tilde{V}_{\pi \circ \nu^*}(s)$
  Initialize array $V(s) \leftarrow 0$ for all $s \in \mathcal{S}$
  **repeat**
    $\Delta \leftarrow 0$
    **for all** $s \in \mathcal{S}$ **do**
      $v \leftarrow \infty, v_0 \leftarrow V(s)$
      **for all** $s_\nu \in B(s)$ **do**
        $v' \leftarrow \sum_{a \in \mathcal{A}} \pi(a|s_\nu) \sum_{s' \in \mathcal{S}} p(s'|s, a) \cdot [R(s, a, s') + \gamma V(s')]$
        $v \leftarrow \min(v, v')$
      **end for**
      $V(s) \leftarrow v$
      $\Delta \leftarrow \max(\Delta, |v_0 - V(s)|)$
    **end for**
  **until** $\Delta < \varepsilon$
  $\tilde{V}_{\pi \circ \nu^*}(s) \leftarrow V(s)$

---

**Theorem 3.** *There exists an SA-MDP and some stochastic policy $\pi \in \Pi_{MR}$ such that we cannot find a better deterministic policy $\pi' \in \Pi_{MD}$ satisfying $\tilde{V}_{\pi' \circ \nu^*(\pi')}(s) \geq \tilde{V}_{\pi \circ \nu^*(\pi)}(s)$ for all $s \in \mathcal{S}$.*

*Proof.* Proof by giving a counter example that no deterministic policy can be better than a random policy. The SA-MDP example in section A provided such a counter example: all 8 possible deterministic policies are no better than the stochastic policy $p_{11} = p_{21} = p_{31} = 0.5$. □

**Theorem 4.** *Under the optimal $\nu^*$, an optimal policy $\pi^* \in \Pi_{MR}$ does not always exist for SA-MDP.*

*Proof.* We will show that the SA-MDP example in section A does not have an optimal policy. First, for $\pi_1$ where $p_{11} = p_{21} = p_{31} = 1$ we have $\tilde{V}_{\pi_1 \circ \nu^*(\pi_1)}(S_1) = 0, \tilde{V}_{\pi_1 \circ \nu^*(\pi_1)}(S_2) = \tilde{V}_{\pi_1 \circ \nu^*(\pi_1)}(S_3) = 100$. This policy is not an optimal policy since we have $\pi_2$ where $p_{11} = p_{21} = p_{31} = 0.5$ that can achieve $\tilde{V}_{\pi_2 \circ \nu^*(\pi_2)}(S_1) = \tilde{V}_{\pi_2 \circ \nu^*(\pi_2)}(S_2) = \tilde{V}_{\pi_2 \circ \nu^*(\pi_2)}(S_3) = 50$ and $\tilde{V}_{\pi_2 \circ \nu^*(\pi_2)}(S_1) > \tilde{V}_{\pi_1 \circ \nu^*(\pi_1)}(S_1)$.

An optimal policy $\pi$, if exists, must be better than $\pi_1$ and have $\tilde{V}_{\pi \circ \nu^*(\pi)}(S_1) > 0, V_{\pi \circ \nu^*(\pi)}(S_2) = V_{\pi \circ \nu^*(\pi)}(S_3) = 100$. In order to achieve $V_{\pi \circ \nu^*(\pi)}(S_2) = V_{\pi \circ \nu^*(\pi)}(S_3) = 100$, we must set $p_{21} = p_{31} = 1$ since it is the only possible way to start from $S_2$ and $S_3$ and receive +1 reward for every step. We can still change $p_{11}$ to probabilities other than 1, however if $p_{11} < 1$ the adversary can set $\nu(S_2) = \nu(S_3) = S_1$ and reduce $V_{\pi \circ \nu^*(\pi)}(S_2)$ and $V_{\pi \circ \nu^*(\pi)}(S_3)$. Thus, no policy better than $\pi_1$ exists, and since $\pi_1$ is not an optimal policy, no optimal policy exists. □

Theorem 3 and Theorem 4 show that the classic definition of optimality is probably not suitable for SA-MDP. Further works can study how to obtain optimal policies for SA-MDP under some alternative definition of optimality, or using a more complex policy class (e.g., history dependent policies).

**Theorem 5.** *Given a policy $\pi$ for a non-adversarial MDP and its value function is $V_\pi(s)$. Under the optimal adversary $\nu$ in SA-MDP, for all $s \in \mathcal{S}$ we have*

$$\max_{s \in \mathcal{S}} \left\{ V_\pi(s) - \tilde{V}_{\pi \circ \nu^*(\pi)}(s) \right\} \leq \alpha \max_{s \in \mathcal{S}} \max_{\hat{s} \in B(s)} \mathrm{D}_{\mathrm{TV}}(\pi(\cdot|s), \pi(\cdot|\hat{s})) \qquad (19)$$

*where $\mathrm{D}_{\mathrm{TV}}(\pi(\cdot|s), \pi(\cdot|\hat{s}))$ is the total variation distance between $\pi(\cdot|s)$ and $\pi(\cdot|\hat{s})$, and $\alpha := 2[1 + \frac{\gamma}{(1-\gamma)^2}] \max_{(s,a,s') \in \mathcal{S} \times \mathcal{A} \times \mathcal{S}} |R(s, a, s')|$ is a constant that does not depend on $\pi$.*

*Proof.* Our proof is based on Theorem 1 in Achiam et al. [1]. In fact, many works in the literature have proved similar results under different scenarios [30, 52]. For an arbitrary starting state $s_0$ and two arbitrary policies $\pi$ and $\pi'$, Theorem 1 in Achiam et al. [1] gives an upper bound of $V_\pi(s_0) - V_{\pi'}(s_0)$.

The bound is given by

$$V_\pi(s_0) - V_{\pi'}(s_0) \le -\mathop{\mathbb{E}}_{\substack{s \sim d_{s_0}^\pi \\ a \sim \pi(\cdot|s) \\ s' \sim p(\cdot|a,s)}} \left[ \left(\frac{\pi'(a|s)}{\pi(a|s)} - 1\right) R(s,a,s') \right]$$

$$+ \frac{2\gamma}{(1-\gamma)^2} \max_s \left\{ \mathop{\mathbb{E}}_{\substack{a \sim \pi'(\cdot|s) \\ s' \sim p(\cdot|a,s)}} \left[ R(s,a,s') \right] \right\} \mathbb{E}_{s \sim d_{s_0}^\pi} \left[ \mathrm{D}_{TV}(\pi(\cdot|s), \pi'(\cdot|s)) \right], \tag{20}$$

where $d_{s_0}^\pi$ is the discounted future state distribution from $s_0$, defined as

$$d_{s_0}^\pi(s) := (1-\gamma) \sum_{t=0}^\infty \gamma^t \mathrm{Pr}(s_t = s | \pi, s_0). \tag{21}$$

Note that in Theorem 1 of Achiam et al. [1], the author proved a general form with an arbitrary function $f$ and we assume $f \equiv 0$ in our proof. We also assume the starting state is deterministic, so $J^\pi$ in Achiam et al. [1] is replaced by $V^\pi(s_0)$. Then we simply need to bound both terms on the right hand side of (20).

For the first term we know that

$$-\mathop{\mathbb{E}}_{\substack{s \sim d_{s_0}^\pi \\ a \sim \pi(\cdot|s) \\ s' \sim p(\cdot|a,s)}} \left[ \left(\frac{\pi'(a|s)}{\pi(a|s)} - 1\right) R(s,a,s') \right] = \sum_s d_{s_0}^\pi(s) \sum_a \left[ \pi(a|s) - \pi'(a|s) \right] \sum_{s'} p(s'|s,a) R(s,a,s')$$

$$\le \sum_s d_{s_0}^\pi(s) \sum_a \left| \pi(a|s) - \pi'(a|s) \right| \left| \sum_{s'} p(s'|s,a) R(s,a,s') \right|$$

$$\le \max_{s,a,s'} |R(s,a,s')| \max_s \left\{ \sum_a \left| \pi(a|s) - \pi'(a|s) \right| \right\}$$

$$= 2 \max_{s,a,s'} |R(s,a,s')| \max_s \mathrm{D}_{TV}(\pi(\cdot|s), \pi'(\cdot|s)) \tag{22}$$

The second term is bounded by

$$\frac{2\gamma}{(1-\gamma)^2} \max_s \left\{ \mathop{\mathbb{E}}_{\substack{a \sim \pi'(\cdot|s) \\ s' \sim p(\cdot|a,s)}} \left[ R(s,a,s') \right] \right\} \mathbb{E}_{s \sim d_{s_0}^\pi} \left[ \mathrm{D}_{TV}(\pi(\cdot|s), \pi'(\cdot|s)) \right]$$

$$\le \frac{2\gamma}{(1-\gamma)^2} \max_{s,a,s'} |R(s,a,s')| \max_s \mathrm{D}_{TV}(\pi(\cdot|s), \pi'(\cdot|s)) \tag{23}$$

Therefore, the RHS of (20) is bounded by $\alpha \max_s \mathrm{D}_{TV}(\pi(\cdot|s), \pi'(\cdot|s))$, where

$$\alpha = 2[1 + \frac{\gamma}{(1-\gamma)^2}] \max_{s,a,s'} |R(s,a,s')| \tag{24}$$

Finally, we simply let $\pi'(\cdot|s) := \pi(\cdot|\nu^*(s))$ and the proof is complete. $\qquad\square$

Before proving Theorem 6 we first give a technical lemma about the total variation distance between two multi-variate Gaussian distributions with the same variance.

**Lemma 2.** *Given two multi-variate Gaussian distributions $X_1 \sim \mathcal{N}(\mu_1, \sigma^2 I_n)$ and $X_2 \sim \mathcal{N}(\mu_2, \sigma^2 I_n)$, $\mu_1, \mu_2 \in \mathbb{R}^n$, define $d = \|\mu_2 - \mu_1\|_2$. We have $\mathrm{D}_{TV}(X_1, X_2) = \sqrt{\frac{2}{\pi}} \frac{d}{\sigma} + O(d^3)$.*

*Proof.* Denote probability density of $X_1$ and $X_2$ as $f_1$ and $f_2$, and denote $a = \frac{\mu_2 - \mu_1}{d}$ as the normal vector of the perpendicular bisector line between $\mu_1$ and $\mu_2$. Due to the symmetry of Gaussian distribution, $f_1(x) - f_2(x)$ is positive for all $x$ where $a^\top x - a^\top \mu_1 - \frac{d}{2} > 0$ and negative for all $x$ on the other symmetric side. When $a^\top x - a^\top \mu_1 - \frac{d}{2} > 0$, $\int_{x \in \mathbb{R}^n} [f_1(x) - f_2(x)] \mathrm{d}x =$

$\Phi(\frac{d}{2\sigma}) - (1 - \Phi(\frac{d}{2\sigma})) = 2\Phi(\frac{d}{2\sigma}) - 1$. Thus,

$$\begin{aligned}
D_{TV}(X_1, X_2) &= \int_{x \in \mathbb{R}^n} |f_1(x) - f_2(x)| \mathrm{d}x \\
&= 2 \int_{a^\top x - a^\top \mu_1 - \frac{d}{2} > 0} (f_1(x) - f_2(x)) \mathrm{d}x \\
&= 2(\Phi(\frac{d}{2\sigma}) - (1 - \Phi(\frac{d}{2\sigma}))) \\
&= 2(2\Phi(\frac{d}{2\sigma}) - 1)
\end{aligned}$$

Then we use the Taylor series for $\Phi(x)$ at $x = 0$:

$$\Phi(x) = \frac{1}{2} + \frac{1}{\sqrt{2\pi}} \sum_{n=0}^{\infty} \frac{(-1)^n x^{2n+1}}{2^n n! (2n+1)}$$

Since we consider the case where $d$ is small, we only keep the first order term and obtain:

$$D_{TV}(X_1, X_2) = \sqrt{\frac{2}{\pi}} \frac{d}{\sigma} + O(d^3)$$

$\square$

**Theorem 6.** $D_{TV}(\bar{\pi}(\cdot|s), \bar{\pi}(\cdot|\hat{s})) = \sqrt{2/\pi} \frac{d}{\sigma} + O(d^3)$, where $d = \|\pi(s) - \pi(\hat{s})\|_2$.

*Proof.* This theorem is a special case of Lemma 2 where $X_1 = \bar{\pi}(\cdot|s)$, $X_2 = \bar{\pi}(\cdot|s')$ and $X_1 \sim \mathcal{N}(\pi(s), \sigma^2 I)$, $X_2 \sim \mathcal{N}(\pi(s'), \sigma^2 I)$. $\square$

## C   Optimization Techniques

### C.1   More Backgrounds for Convex Relaxation of Neural Networks

In our work, we frequently need to solve a minimax problem:

$$\min_{\theta} \max_{\phi \in \mathbb{S}} g(\theta, \phi) \tag{25}$$

One approach we will discuss is to first solve the inner maximization problem (approximately) using an optimizer like SGLD. However, due to the non-convexity of $\pi_\theta$, we cannot solve the inner maximization to global maxima, and the gap between local maxima and global maxima can be large. Using convex relaxations of neural networks, we can instead find an upper bound of $\max_{\phi \in \mathbb{S}} g(\theta, \phi)$:

$$\bar{g}(\theta) \geq \max_{\phi \in \mathbb{S}} g(\theta, \phi)$$

Thus we can minimize an upper bound instead, which can guarantee the original objective (25) is minimized.

As an illustration on how to find $\bar{g}(\theta)$ using convex relaxations, following Salman et al. [57] we consider a simple $L$-layer MLP network $f(\theta, x)$ with parameters $\theta = \{(W^{(i)}, b^{(i)}), i \in \{1, \cdots, L\}\}$ and activation function $\sigma$. We denote $x^{(0)} = x$ as the input, $x^{(i)}$ as the post-activation value for layer $i$, $z^{(i)}$ as the pre-activation value for layer $i$. $i \in \{1, \cdots, L\}$. The output of the network $f(\theta, x)$ is $z^{(L)}$. Then, we consider the following optimization problem:

$$\max_{x \in \mathbb{S}} f(\theta, x), \quad \text{where } \mathbb{S} \text{ is the set of perturbations}$$

which is equivalent to the following optimization problem:

$$\begin{aligned}
\max \quad & z^{(L)} \\
\text{s.t.} \quad & z^{(l)} = W^{(l)} x^{(l-1)} + b^{(l)}, l \in [L], \\
& x^{(l)} = \sigma(z^{(l)}), l \in [L-1], \\
& x^{(0)} \in \mathbb{S}
\end{aligned} \tag{26}$$

In this constrained optimization problem (26), assuming $\mathbb{S}$ is a convex set, the constraint on $z^{(l)}$ is convex (linear) and the only non-convex constraints are those for $x^{(l)}, l = \{1, \cdots, L-1\}$, where a non-linear activation function is involved. Note that activation function $\sigma(z)$ itself can be a convex function, but when used as an equality constraint, the feasible solution is constrained to the *graph* of $\sigma(z)$, which is non-convex.

Previous works [79, 86, 57] propose to use convex relaxations of non-linear units to relax the non-convex constraint $x^{(l)} = \sigma(z^{(l)})$ with a convex one, $x^{(l)} = \text{convex}(\sigma(z^{(l)}))$, such that (26) can be solved efficiently. We can then obtain an *upper bound* of $f(\theta, x)$ since the constraints are relaxed.

Zhang et al. [86] gave several concrete examples (e.g., ReLU, tanh, sigmoid) on how these relaxations are formed. In the special case where linear relaxations are used, (26) can be solved efficiently and automatically (without manual derivation and implementation) for general computational graphs [83]. Generally, using the framework from Xu et al. [83] we can access an oracle function ConvexRelaxUB defined as below:

**Definition 2.** *Given a neural network function $f(\mathbf{X})$ where $\mathbf{X}$ is any input for this function, and $\mathbf{X} \in \mathbb{S}$ where $\mathbb{S}$ is the set of perturbations, the oracle function* ConvexRelaxUB *provided by an automatic neural network convex relaxation tool returns an upper bound $\overline{f}$, which satisfies:*

$$\overline{f} \geq \max_{\mathbf{X} \in \mathbb{S}} f(\mathbf{X})$$

Note that in the above definition, $\mathbf{X}$ can by *any* input for this computation (e.g., $\mathbf{X}$ can be $s$, $a$, or $\theta$ for a $Q_\theta(s, a)$ function). In the special case of our paper, for simplicity we define the notation ConvexRelaxUB$(f, \theta, s \in B(s))$ which returns an upper bound function $\overline{f}(\theta)$ for $\max_{s \in B(s)} f(\theta, s)$.

**Computational cost**  Many kinds of convex relaxation based methods exist [57], where the expensive ones (which give a tighter upper bound) can be a few magnitudes slower than forward propagation. The cheapest method is interval bound propagation (IBP), which only incurs twice more costs as forward propagation; however, IBP base training has been reported unstable and hard to reproduce as its bounds are very loose [88, 3]. To avoid potential issues with IBP, in all our environments, we use the IBP+Backward relaxation scheme following [88, 83], which produces considerably tighter bounds, while being only a few times slower than forward propagation (e.g., 3 times slower than forward propagation when loss fusion [83] is implemented). In fact, Xu et al. [83] used the same relaxation for training downscaled ImageNet dataset on very large vision models. For DRL the policy neural networks are typically small and can be handled quite efficiently. In our paper, we use convex relaxation as a blackbox tool (provided by the `auto_LiRPA` library [83]), and any new development for improving its efficiency can benefit us.

### C.2  Solving the Robust Policy Regularizer using SGLD

Stochastic gradient Langevin dynamics (SGLD) [18] can escape saddle points and shallow local optima in non-convex optimization problems [54, 89, 10, 84], and can be used to solve the inner maximization with zero gradient at $\hat{s} = s$. SGLD uses the following update rule to find $\hat{s}^K$ to maximize $\mathcal{R}_s(\hat{s}, \theta_\mu)$:

$$\hat{s}^{k+1} \leftarrow \text{proj}\left(\hat{s}^k - \eta_k \nabla_{\hat{s}^k} \mathcal{R}_s(\hat{s}^k, \theta_\mu) + \sqrt{2\eta_k/\beta_k}\xi\right), \quad \hat{s}^1 = s, \quad k = 1, \cdots, K$$

where $\eta_k$ is step size, $\xi$ is an i.i.d. standard Gaussian random variable in $\mathbb{R}^{|\mathcal{S}|}$, $\beta_k$ is an inverse temperature hyperparameter, and proj$(\cdot)$ projects the update back into $B(s)$. We find that SGLD is sufficient to escape the stationary point at $\hat{s} = s$. However, due to the non-convexity of $\mu_{\theta_\mu}(\hat{s}, \theta_\mu)$, this approach only provides a lower bound $\mathcal{R}_s(\hat{s}^K, \theta_\mu)$ of $\max_{\hat{s} \in B(s)} \mathcal{R}_s(\hat{s}, \theta_\mu)$. Unlike the convex relaxation based approach, minimizing this lower bound does not guarantee to minimize (5), as the gap between $\max_{\hat{s} \in B(s)} \mathcal{R}_s(\hat{s}, \theta_\mu)$ and $\mathcal{R}_s(\hat{s}^K, \theta_\mu)$ can be large.

**Computational Cost**  In SGLD, we first need to solve the inner maximization problem (such as Eq. (5)). The additional time cost depends on the number of SGLD steps. In our experiments for PPO and DDPG, we find that using 10 steps are sufficient. However, the total training cost does not grow by 10 times, as in many environments the majority of time was spent on environment simulation steps, rather than optimizing a small policy network.

# D  Additional details for adversarial attacks on state observations

## D.1  More details on the Critic based attack

In Section 3.5 we discuss the critic based attack [50] as a baseline. This attack requires a $Q$ function $Q(s, a)$ to find the best perturbed state. In Algorithm 2 we present our "corrected" critic based attack based on [50]:

---

**Algorithm 2** Critic based attack [50]

---

**Input:** A policy function $\pi$ under attack, a corresponding $Q(s, a)$ network, and a initial state $s_0$, $T$ is the number of attack steps, $\eta$ is the step size, $\underline{s}$ and $\overline{s}$ are valid lower and upper range of $s$ (assuming a $\ell_\infty$ norm-like threat model).
    **for** $t = 1$ to $T$ **do**
        $g_t = \nabla Q_{s_{t-1}}(s_0, \pi(s_{t-1})) = \frac{\partial Q}{\partial \pi}\frac{\partial \pi}{\partial s_{t-1}}$
        $g_t \leftarrow \text{proj}(g_t)$ ▷project $g_t$ according to norm constraint of $s$; for $\ell_\infty$ norm simply take the sign
        $s_t \leftarrow s_{t-1} - \eta g_t$
        $s_t \leftarrow \min(\max(s_t, \underline{s}), \overline{s})$                        ▷only needed for $\ell_\infty$ norm threat model
    **end for**
**Output:** An adversarial state $\hat{s} := s_T$

---

Note that in Algorithm 4 of [50], they use the gradient $\nabla Q_s(s, \pi(s)) = \frac{\partial Q}{\partial s} + \frac{\partial Q}{\partial \pi}\frac{\partial \pi}{\partial s}$ which essentially attempts to minimize $Q(\hat{s}, \pi(\hat{s}))$, but they then sample randomly along this gradient direction to find the best $\hat{s}$ that minimizes $Q(s_0, \pi(\hat{s}))$. Our corrected formulation directly minimizes $Q(s_0, \pi(\hat{s}))$ using this gradient instead $\nabla Q_s(s_0, \pi(s)) = \frac{\partial Q}{\partial \pi}\frac{\partial \pi}{\partial s}$.

For PPO, since there is no $Q(s, a)$ available during training, we extend [50] to perform attack relying on $V(s)$: we find a state $\hat{s}$ that minimizes $V(\hat{s})$. Unfortunately, it does not match our setting of perturbing state observations; it looks for a state $\hat{s}$ that has the worst value (i.e., taking action $\pi(\hat{s})$ in state $\hat{s}$ is bad), but taking the action $\pi(\hat{s})$ at state $s_0$ does not necessarily trigger a low reward action, because $V(\hat{s}) = \max_a Q(\hat{s}, a) \neq \max_a Q(s_0, a)$. Thus, in Table 1 we can observe that critic based attack typically does not work very well for PPO agents.

## D.2  More details on the Maximal Action Difference (MAD) attack

We present the full algorithm of MAD attack in Algorithm 3. It is a relatively simple attack by directly maximizing a KL-divergence using SGLD, yet it usually outperforms random attack and critic attack on some environments (e.g., see Figure 10).

---

**Algorithm 3** Maximal Action Difference (MAD) Attack (a critic-independent attack)

---

**Input:** A policy function $\pi$ under attack, and a initial state $s_0$, $T$ is the number of attack steps, $\eta$ is the step size, $\beta$ is the (inverse) temperature parameter for SGLD, $\underline{s}$ and $\overline{s}$ are valid lower and upper range of $s$.
    Define loss function $L_{\text{MAD}}(s) = -D_{\text{KL}}(\pi(\cdot|s_0)\|\pi(\cdot|s))$
    **for** $t = 1$ to $T$ **do**
        Sample $\xi \sim \mathcal{N}(0, 1)$
        $g_t = \nabla L_{\text{MAD}}(s_{t-1}) + \sqrt{\frac{2}{\beta\eta}}\xi$
        $g_t \leftarrow \text{proj}(g_t)$ ▷project $g_t$ according to norm constraint of $s$; for $\ell_\infty$ norm simply take the sign
        $s_t \leftarrow s_{t-1} - \eta g_t$
        $s_t \leftarrow \min(\max(s_t, \underline{s}), \overline{s})$
    **end for**
**Output:** An adversarial state $\hat{s} := s_T$

---

## D.3  More details on the Robust Sarsa attack

Algorithm 4 gives the full procedure of the Robust Sarsa attack. We collect trajectories of the agents and then optimize the ordinary temporal difference (TD) loss along with a robust objective $L_{\text{robust}}(\theta)$. $L_{\text{robust}}(\theta)$ constrains that when an input action $a$ is slightly changed, the value $Q_{\text{RS}}^\pi(s, a)$ should not

change significantly. We set the perturbation set $B_p(a, \epsilon)$ to be a $\ell_p$ norm ball with radius $\epsilon$ around an action $a$. We gradually increase $\epsilon$ from 0 to $\epsilon_{\max}$ during training to learn a critic that is increasingly more robust. The inner maximization of $L_{\mathrm{robust}}(\theta)$ is upper bounded by convex relaxations of neural networks, which we introduced in section C.1. Once the inner maximization is eliminated, we solve the final objective using regular first order optimization methods. In our attacks to DDPG and PPO, we try multiple regularization parameter $\lambda_{\mathrm{RS}}$ to find the best Sarsa model that achieves *lowest* attack rewards.

---

**Algorithm 4** Train a robust value function for critic-independent attack (Robust Sarsa attack)

---

**Input:** Any policy function $\pi$ under attack, $T$ is the number of training steps, and an epsilon schedule $\epsilon_t$

Initialize $Q_{\mathrm{RS}}^\pi(s, a)$ to be a random network
**for** $t = 1$ to $T$ **do**
    Run the agent with policy $\pi$ and collect a batch of $N$ steps: $\{s_i, a_i, r_i, s_i', a_i'\}, i \in [N]$
    $L_{\mathrm{TD}}(\theta) = \sum_{i \in [N]} \left[ r_i + \gamma Q_{\mathrm{RS}}^\pi(s_i', a_i') - Q_{\mathrm{RS}}^\pi(s_i, a_i) \right]^2$
    $L_{\mathrm{robust}}(\theta) = \sum_{i \in [N]} \max_{\hat{a} \in B_p(a_i, \epsilon_t)} (Q_{\mathrm{RS}}^\pi(s_i, \hat{a}) - Q_{\mathrm{RS}}^\pi(s_i, a_i))^2$
    $\overline{L}_{\mathrm{robust}} = \mathrm{ConvexRelaxUB}(L_{\mathrm{robust}}, \theta, B_p(a_i, \epsilon_t))$, where $L_{\mathrm{robust}}(\theta) \leq \overline{L}_{\mathrm{robust}}(\theta)$   ▷Solving the inner maximization by upper bounding $L_{\mathrm{robust}}$ using an automatic NN convex relaxation tool
    Minimize $L_{\mathrm{RS}}(\theta) = L_{\mathrm{TD}}(\theta) + \lambda_{\mathrm{RS}} \overline{L}_{\mathrm{robust}}(\theta)$ using any gradient based optimizer (e.g., Adam)
**end for**
**Output:** A robust critic function $Q_{\mathrm{RS}}^\pi$ that can be used for Algorithm 2.

---

Although it is beyond the scope of this paper, RS attack can also be used as a blackbox attack when perturbing the actions rather than state observations, as $Q_{\theta_{RS}}^\pi$ can be learned by observing the environment and the agent without any internal information of the agent. Then, using the robust critic we learned, black-box attacks can be performed on action space by solving $\min Q_{\theta_{RS}}^\pi(s, a)$ with a norm constrained $a$.

For a practical implementation, to improve convergence and reduce instability, two $Q_{\mathrm{RS}}^\pi(s, a)$ functions can be also used similarly as in double Q learning [23]. In our case, since the policy is not being updated and stable, we find that using a single Q function is also sufficient for most settings and usually converges faster.

We provide some empirical justifications for the necessity of using a robust objective. For both PPO and DDPG, we conduct attacks using a Sarsa network trained with and without the robustness objective, in Table 4 and Table 5, respectively. We observe that the robust objective can decrease reward further more in most settings.

Table 4: Comparison between Non-robust Sarsa attack (without the robustness objective $L_{\mathrm{robust}}(\theta)$) and robust Sarsa attack on PPO and SA-PPO agents in Table 1. The Robust Sarsa Attack Reward column is the same result presented in RS column of Table 1. We report mean reward $\pm$ standard deviation over 50 attack episodes.

| Env. | $\ell_\infty$ norm perturbation budget $\epsilon$ | Method | Non-robust Sarsa Attack Reward | Robust Sarsa Attack Reward |
|---|---|---|---|---|
| Hopper | 0.05 | PPO (vanilla) | 2757.0±604.2 | **779.4±33.2** |
| | | PPO (adv. 50%) | 276 ±140 | **49 ± 50** |
| | | PPO (adv. 100%) | 14.4± 4.20 | **3.8 ± 0.9** |
| | | SA-PPO (SGLD) | 3642.9±4.0 | **1403.3±55.0** |
| | | SA-PPO (Convex) | 3014.9±656.1 | **1235.8±50.2** |
| Walker2d | 0.05 | PPO (vanilla) | 2224.7±1438.7 | **913.7±54.3** |
| | | PPO (adv. 50%) | -10.79 ± 0.93 | **-11.55 ± 0.79** |
| | | PPO (adv. 100%) | -111.9± 4.5 | **-114.4 ± 4.0** |
| | | SA-PPO (SGLD) | 4777.1±305.5 | **2605.6±1255.7** |
| | | SA-PPO (Convex) | 3701.1±1013.3 | **2168.2± 665.4** |
| Humanoid | 0.075 | PPO (vanilla) | **716.4±166.1** | 1036.0±420.2 |
| | | PPO (adv. 50%) | 166± 78 | **98 ± 69** |
| | | PPO (adv. 100%) | 122.6± 15.9 | **113.2 ± 18.5** |
| | | SA-PPO (SGLD) | 6115.4±783.2 | **6200.5±818.1** |
| | | SA-PPO (Convex) | 6241.2±540.8 | **4707.2±1359.1** |

Table 5: Comparison between Non-robust Sarsa attack (without the robustness objective) and robust Sarsa attack on DDPG and SA-DDPG agents in Table 2. The Robust Sarsa Attack Reward column presents the same results as presented in the RS attack rows of Table 6. We report mean reward $\pm$ standard deviation over 50 attack episodes.

| Env. | $\ell_\infty$ norm perturbation budget $\epsilon$ | Method | Non-robust Sarsa Attack Reward | Robust Sarsa Attack Reward |
|---|---|---|---|---|
| Ant | 0.2 | DDPG (vanilla) | $700 \pm 305$ | $\mathbf{577 \pm 394}$ |
| | | SA-DDPG (Convex) | $2380 \pm 142$ | $\mathbf{2276 \pm 242}$ |
| Hopper | 0.075 | DDPG (vanilla) | $1362 \pm 1468$ | $\mathbf{606 \pm 124}$ |
| | | SA-DDPG (Convex) | $1323 \pm 491$ | $\mathbf{1258 \pm 561}$ |
| InvertedPendulum | 0.3 | DDPG (vanilla) | $1000 \pm 0$ | $\mathbf{92 \pm 1}$ |
| | | SA-DDPG (Convex) | $1000 \pm 0$ | $1000 \pm 0$ |
| Reacher | 1.5 | DDPG (vanilla) | $-24.11 \pm 7.19$ | $\mathbf{-21.74 \pm 5.14}$ |
| | | SA-DDPG (Convex) | $-11.67 \pm 3.57$ | $-11.40 \pm 3.56$ |
| Walker2d | 0.05 | DDPG (vanilla) | $951 \pm 1146$ | $959 \pm 1001$ |
| | | SA-DDPG (Convex) | $3200 \pm 1939$ | $\mathbf{1986 \pm 1993}$ |

### D.4  Hybrid RS+MAD attack

We find that RS and MAD attack can achieve the best results (lowest attack reward) in many cases. We also consider combining them to form a hybrid attack, which minimizes the robust critic predicted value and in the meanwhile maximizes action differences. It can be conducted by minimizing this combined loss function to find an adversarial state $\hat{s} \in B(s)$:

$$L_{\text{Hybrid}}(\hat{s}) = \alpha_{\text{RS-MAD}} Q_{\theta_Q}(s, \pi_{\theta_{RS}}(\hat{s})) + (1 - \alpha_{\text{RS-MAD}}) L_{\text{MAD}}(\hat{s})$$

For a practical implementation, it is important to choose $\alpha_{\text{RS-MAD}}$ so that the two parts of the loss are roughly balanced. The value of $Q_{\theta_Q}$ depends on environment reward (if reward is not normalized), and might be much larger in magnitudes than RS-MAD, so typically $\alpha_{\text{RS-MAD}}$ is close to 1.

We try different values of $\alpha_{\text{RS-MAD}}$ and report the lowest reward as the final reward under this attack.

### D.5  Projected Gradient Decent (PGD) Attack for DQN

For DQN, we use the regular untargeted Projected Gradient Decent (PGD) attack in the literature [36, 50, 81]. The untargeted PGD attack with $K$ iterations updates the state $K$ times as follows:

$$\begin{aligned} s^{k+1} &= s^k + \eta \text{proj}[\nabla_{s^k} \mathcal{H}(Q_\theta(s^k, \cdot), a^*)], \\ s^0 &= s, \quad k = 0, \dots, K-1 \end{aligned} \tag{27}$$

where $\mathcal{H}(Q_\theta(s^k, \cdot), a^*)$ is the cross-entropy loss between the output logits of $Q_\theta(s^k, \cdot)$ and the onehot-encoded distribution of $a^* := \arg\max_a Q_\theta(s, a)$. proj$[\cdot]$ is a projection operator depending on the norm constraint of $B(s)$ and $\eta$ is the learning rate. A successful untargeted PGD attack will then perturb the state to lead the Q network to output an action other than the optimal action $a^*$ chosen at the original state $s$. To guarantee that the final state obtained by the attack is within an $\ell_\infty$ ball around $s$ ($B_\epsilon(s) = \{\hat{s} : s - \epsilon \leq \hat{s} \leq s + \epsilon\}$), the projection proj$[\cdot]$ is a sign operator and $\eta$ is typically set to $\eta = \frac{\epsilon}{K}$.

## E  Robustness Certificates for Deep Reinforcement Learning

If we use the convex relaxation in Section C.1 to train our networks, it can produce robustness certificates [79, 44, 88] for our task. However in some RL tasks the certificates have interpretations different from classification tasks, as discussed in detail below.

**Robustness Certificates for DQN.** In DQN, the action space is finite, so we have a robustness certificate on the actions taken at each state. More specifically, at each state $s$, policy $\pi$'s action is certified if its corresponding Q function satisfies

$$\arg\max_a Q_\theta(s, a) = \arg\max_a Q_\theta(\hat{s}, a) = a^*, \text{for all } \hat{s} \in B(s). \tag{28}$$

Given a states $s$, we can use neural network convex relaxations to compute an upper bound $u_{Q_\theta, a^*, a}(s)$ such that

$$Q_\theta(\hat{s}, a) - Q_\theta(\hat{s}, a^*) \leq u_{Q_\theta, a^*, a}(s)$$

holds for all $\hat{s} \in B(s)$. So if $u_{Q_\theta, a^*, a}(s) \leq 0$ for all $a \in \mathcal{A}$, we have

$$Q_\theta(\hat{s}, a) - Q_\theta(\hat{s}, a^*) \leq 0 \tag{29}$$

is guaranteed for all $\hat{s} \in B(s)$, which means that the agent's action will not change when the state observation is in $B(s)$. When the agent's action is not changed under an adversarial perturbation, its reward and transition at current step will not change in the DQN setting, either.

In some settings, we find that 100% of the actions are guaranteed to be unchanged (e.g., the Pong environment in Table 3). In that case, we can in fact also certify that the accumulated reward is not changed given the specific initial conditions for testing. Otherwise, if some steps during the roll-out do not have this certificate, or have a weaker certificate that more than one actions are possible given $\hat{s} \in B(s)$, all the possible actions have to be explored as the next action input to the environment. When there are $n$ states which are not certified to have unchanged actions, each with $m$ possible actions, we need to run $n^m$ trajectories to find the worst case cumulative reward. This is impractical for typical settings.

However, even in the 100% certificate rate setting like Pong, it can still be challenging to certify that the agent is robust under *any* starting condition. Since the agent is started with a random initialization, it is impractical to enumerate all possible initializations and guarantee all generated trajectories are certified. Similarly, in the classification setting, many existing certified defenses [80, 44, 19, 88] can only practically guarantee robustness on a specific test set (by computing a "verified test error"), rather than on *any* input image.

**Robustness Certificates for PPO and DDPG.** In DDPG and PPO, the action space is continuous, hence it is not possible to certify that actions do not change under adversary. We instead seek for a different type of guarantee, where we can upper bound the change in action given a norm bounded input perturbation:

$$U_s \geq \max_{\hat{s} \in B(s)} \|\pi_{\theta_\pi}(\hat{s}) - \pi_{\theta_\pi}(s)\| \tag{30}$$

Given a state $s$, we can use convex relaxations to compute an upper bound $U_s$. Generally speaking, if $B(s)$ is small, a robust policy desires to have a small $U_s$, otherwise it can be possible to find an adversarial state perturbation that greatly changes $\pi_{\theta_\pi}(\hat{s})$ and causes the agent to misbehave. However, giving certificates on cumulative rewards is still challenging, as it requires to bound reward $r(s, a)$ given a fixed state $s$, and a perturbed and bounded action $a$ (bounded via (30)). Since the environment dynamics can be quite complex in practice (except for the simplest environment like InvertedPendulum), it is hard to bound reward changes given a bounded action. We leave this part as a future direction for exploration and we believe the robustness certificates in (30) can be useful for future works.

# F   Additional details for SA-PPO

**Algorithm**   We present the full SA-PPO algorithm in Algorithm 5. Compared to vanilla PPO, we add a robust state-adversarial regularizer which constrains the KL divergence on state perturbations. We highlighted these changes in Algorithm 5. The regularizer $\mathcal{R}_{\text{PPO}}(\theta_\pi)$ can be solved using SGLD or convex relaxations of neural networks. We define the perturbation set $B(s)$ to be an $\ell_p$ norm ball around state $s$ with radius $\epsilon$: $B_p(s, \epsilon) := \{s' | \|s' - s\|_p \leq \epsilon\}$. We use a $\epsilon$-schedule during training, where the perturbation budget is slowly increasing dduring each epoch $t$ as $\epsilon_t$ until reaching $\epsilon$.

**Hyperparameters for Regular PPO Training**   We use the optimal hyperparameters in [14] which were found using a grid search for vanilla PPO. However, we found that their parameters are not optimal for Humanoid and achieves a cumulative reward of only about 2000 after $1 \times 10^7$ steps. Thus we redo hyperparameter search on Humanoid and change learning rate for actor to $5 \times 10^{-5}$ and critic to $1 \times 10^{-5}$. This new set of hyeremeters allows us to obtain Humanoid reward about 5000 for vanilla PPO. Note that even under the original, non-optimal set of hyperemeters by [14], our SA-PPO variants still achieve high rewards similarly to those reported in our paper. Our hyperparameter change only significantly improves the performance of vanilla PPO baseline.

We run 2048 simulation steps per iteration, and run policy optimization of 10 epochs with a minibatch size of 64 using Adam optimizer with learning rate $3 \times 10^{-4}$, $4 \times 10^{-4}$ and $5 \times 10^{-5}$ for Walker, Hopper and Humanoid, respectively. The value network is also trained in 10 epochs per iteration with a minibatch size of 64, using Adam optimizer with learning rate 0.00025, $3 \times 10^{-4}$, and $1 \times 10^{-5}$ for Walker, Hopper and Humanoid environments, respectively (the same as in [14] without further tuning, except for Humanoid as discussed above). Both networks are 3-layer MLPs with $[64, 64]$

**Algorithm 5** State-Adversarial Proximal Policy Optimization (SA-PPO)

**Input:** Number of iterations $T$, a $\epsilon$ schedule $\epsilon_t$

1: Initialize actor network $\pi(a|s)$ and critic network $V(s)$ with parameter $\theta_\pi$ and $\theta_V$,
2: **for** $t = 1$ to $T$ **do**
3:  Run $\pi_{\theta_\pi}$ to collect a set of trajectories $\mathcal{D} = \{\tau_k\}$ containing $|\mathcal{D}|$ episodes, each $\tau_k$ is a trajectory contain $|\tau_k|$ samples, $\tau_k := \{(s_{k,i}, a_{k,i}, r_{k,i}, s_{k,i+1})\}, i \in [|\tau_k|]$
4:  Compute cumulative reward $\hat{R}_{k,i}$ for each step $i$ in every episode $k$ using the trajectories and discount factor $\gamma$
5:  Update Value function by minimizing the mean-square error:

$$\theta_V \leftarrow \arg\min_{\theta_V} \frac{1}{\sum_k |\tau_k|} \sum_{\tau_k \in D} \sum_{i=0}^{|\tau_k|} \left( V(s_{k,i}) - \hat{R}_{k,i} \right)^2$$

6:  Estimate advantage $\hat{A}_{k,i}$ for each step $i$ in every episode $k$ using generalized advantage estimation (GAE) and value function $V_{\theta_V}(s)$
7:  Define the state-adversarial policy regularier:

$$\mathcal{R}_{\text{PPO}}(\theta_\pi) := \sum_{\tau_k \in D} \sum_{i=0}^{|\tau_k|} \max_{\bar{s}_{k,i} \in B_p(s_{k,i}, \epsilon_t)} \mathrm{D}_{\mathrm{KL}}\left(\pi(a|s_{k,i}) \| \pi(a|\bar{s}_{k,i})\right)$$

8:  Option 1: Solve $\mathcal{R}_{\text{PPO}}(\theta_\pi)$ using SGLD:
9:   find $\hat{s}_{k,i} = \arg\max_{\bar{s}_{k,i} \in B_p(s_{k,i}, \epsilon_t)} \mathrm{D}_{\mathrm{KL}}(\pi(a|s_{k,i}) \| \pi(a|\bar{s}_{k,i}))$ using SGLD optimization for all $k, i$ (the objective can be solved in a batch)
10:   set $\overline{\mathcal{R}}_{\text{PPO}}(\theta_\pi) := \sum_{\tau_k \in D} \sum_{i=0}^{|\tau_k|} \mathrm{D}_{\mathrm{KL}}(\pi(a|s_{k,i}) \| \pi(a|\hat{s}_{k,i}))$
11:  Option 2: Solve $\mathcal{R}_{\text{PPO}}(\theta_\pi)$ using convex relaxations:
12:   $\overline{\mathcal{R}}_{\text{PPO}}(\theta_\pi) := \text{ConvexRelaxUB}(\mathcal{R}_{\text{PPO}}, \theta_\pi, \bar{s}_{k,i} \in B_p(s_{k,i}, \epsilon_t))$
13:  Update the policy by minimizing the SA-PPO objective (the minimization is solved using ADAM):

$$\theta_\pi \leftarrow \arg\min_{\theta'_\pi} \frac{1}{\sum_k |\tau_k|} \left[ \sum_{\tau_k \in D} \sum_{i=0}^{|\tau_k|} \min\left( r_{\theta'_\pi}(a_{k,i}|s_{k,i})\hat{A}_{k,i}, g(r_{\theta'_\pi}(a_{k,i}|s_{k,i}))\hat{A}_{k,i} \right) + \kappa_{\text{PPO}}\overline{\mathcal{R}}_{\text{PPO}}(\theta'_\pi) \right]$$

where $r_{\theta'_\pi}(a_{k,i}|s_{k,i}) := \frac{\pi_{\theta'_\pi}(a_{k,i}|s_{k,i})}{\pi_{\theta_\pi}(a_{k,i}|s_{k,i})}, g(r) := \text{clip}(r_{\theta'_\pi}(a_{k,i}|s_{k,i}), 1 - \epsilon_{\text{clip}}, 1 + \epsilon_{\text{clip}})$

14: **end for**

---

hidden neurons. The clipping value $\epsilon$ for PPO is 0.2. We clip rewards to $[-10, 10]$ and states to $[-10, 10]$. The discount factor $\gamma$ for reward is 0.99 and the discount factor used in generalized advantage estimation (GAE) is 0.95. We found that in [14] the agent rewards are still improving when training finishes, thus in our experiments we run the agents longer for better convergence: we run Walker2d and Hopper $2 \times 10^6$ steps (976 iterations) and Humanoid $1 \times 10^7$ steps (4882 iterations) to ensure convergence.

**Hyperparameter for SA-PPO Training** For SA-PPO, we use the same set of hyperparameters as in PPO. Note that the hyperparameters are tuned for PPO but not specifically for SA-PPO. The additional regularization parameter $\kappa_{\text{PPO}}$ for the regularizer $\mathcal{R}_{\text{PPO}}$ is chosen in $\{0.003, 0.01, 0.03, 0.1, 0.3, 1.0\}$. We linearly increase $\epsilon_t$, the norm of $\ell_\infty$ perturbation on normalized states, from 0 to the target value ($\epsilon$ for evaluation, reported in Table 1) during the first $3/4$ iterations, and keep $\epsilon_t = \epsilon$ for the reset iterations. The same $\epsilon$ schedule is used for both SGLD and convex relaxation training. For SGLD, we run 10 iterations with step size $\frac{\epsilon_t}{10}$ and set the temperature parameter $\beta = 1 \times 10^{-5}$. For convex relaxations, we use the efficient IBP+Backward scheme [83], and we use a training schedule similar to [88] by mixing the IBP bounds and backward mode perturbation analysis bounds.

# G   Additional Details for SA-DDPG

**Algorithm**   We present the SA-DDPG training algorithm in Algorithm 6. The main difference between DDPG and SA-DDPG is the additional loss term $\mathcal{R}_{\text{DDPG}}(\theta_\pi)$, which provides an upper bound on $\max_{s \in B(s_i)} \|\pi(s) - \pi(s_i)\|_2^2$. We highlighted these changes in Algorithm 6. We define the perturbation set $B(s)$ to be a $\ell_p$ norm ball around $s$ with radius $\epsilon$: $B_p(s, \epsilon) := \{s' | \|s' - s\|_p \le \epsilon\}$. We use a $\epsilon$-schedule during training, where the perturbation budget is slowly increasing during training as $\epsilon_t$ until reaching $\epsilon$.

---

**Algorithm 6** State-Adversarial Deep Deterministic Policy Gradient (SA-DDPG)

---

Initialize actor network $\pi(s)$ and critic network $Q(s, a)$ with parameter $\theta_\pi$ and $\theta_Q$
Initialize target network $\pi'(s)$ and critic network $Q'(s, a)$ with weights $\theta_{\pi'} \leftarrow \theta_\pi$ and $\theta_{Q'} \leftarrow \theta_Q$
Initial replay buffer $\mathcal{B}$
**for** $t = 1$ to $T$ **do**
   Initial a random process $\mathcal{N}$ for action exploration
   Choose action $a_t \sim \pi(s_t) + \epsilon, \epsilon \sim \mathcal{N}$
   Observe reward $r_t$, next state $s_{t+1}$ from environment
   Store transition $\{s_t, a_t, r_t, s_{t+1}\}$ into $\mathcal{B}$
   Sample a mini-batch of $N$ samples $\{s_i, a_i, r_i, s_i'\}$ from $\mathcal{B}$
   $y_i \leftarrow r_i + \gamma Q'(s_i', \pi'(s_i'))$ for all $i \in [N]$
   Update $\theta_Q$ by minimizing loss $L(\theta_Q) = \frac{1}{N} \sum_i (y_i - Q(s_i, a_i))^2$
   $\mathcal{R}_{\text{DDPG}}(\theta_\pi, \bar{s}_i) := \sum_i \max_{\bar{s}_i \in B_p(s_i, \epsilon_t)} \|\pi_{\theta_\pi}(s_i) - \pi_{\theta_\pi}(\bar{s}_i)\|_2$
   Option 1: Solve $\mathcal{R}_{\text{DDPG}}(\theta_\pi)$ using SGLD:
      find $\hat{s}_i = \arg\max_{\bar{s}_i \in B_p(s_i, \epsilon_t)} \|\pi_{\theta_\pi}(s_i) - \pi_{\theta_\pi}(\bar{s}_i)\|_2$ for all $i$ (solved in a batch using SGLD)
      set $\overline{\mathcal{R}}_{\text{DDPG}}(\theta_\pi) := \sum_i \|\pi_{\theta_\pi}(s_i) - \pi_{\theta_\pi}(\hat{s}_i)\|_2$
   Option 2: Solve $\mathcal{R}_{\text{DDPG}}(\theta_\pi)$ using convex relaxations:
      $\overline{\mathcal{R}}_{\text{DDPG}}(\theta_\pi) := \text{ConvexRelaxUB}(\mathcal{R}_{\text{DDPG}}, \theta_\pi, \bar{s}_i \in B_p(s_i, \epsilon_t))$
   Update $\theta_\pi$ using deterministic policy gradient and gradient of $\overline{\mathcal{R}}_{\text{DDPG}}$:
   $\nabla_{\theta_\pi} J(\theta_\pi) = \frac{1}{N} \sum_i \left[ \nabla_a Q(s, a)|_{s=s_i, a=\pi(s_i)} \nabla_{\theta_\pi} \pi(s)|_{s=s_i} + \kappa_{\text{DDPG}} \nabla_{\theta_\pi} \overline{\mathcal{R}}_{\text{DDPG}} \right]$
   Update Target Network:
   $\theta_{Q'} \leftarrow \tau \theta_Q + (1 - \tau)\theta_{Q'}$
   $\theta_{\pi'} \leftarrow \tau \theta_\pi + (1 - \tau)\theta_{\pi'}$
**end for**

---

**Hyperparameters for Regular DDPG Training.**   Our hyperparameters are from [61]. Both actor and critic networks are 3-layer MLPs with $[400, 300]$ hidden neurons. We run each environment for $2 \times 10^6$ steps. Actor network learning rate is $1 \times 10^{-4}$ and critic network learning rate is $1 \times 10^{-3}$ (except that for Hopper-v2 and Ant-v2 the critic learning rate is reduced to $1 \times 10^{-4}$ due to the larger values of rewards); both networks are optimized using Adam optimizer. No reward scaling is used, and discount factor is set to 0.99. We use a replay buffer with a capacity of $1 \times 10^6$ items and we do not use prioritized replay buffer sampling. For the random process $\mathcal{N}$ used for exploration, we use a Ornstein-Uhlenbeck process with $\theta = 0.15$ and $\sigma = 0.2$. The mixing parameter of current and target actor and critic networks is set to $\tau = 0.001$.

**Hyperparameters for SA-DDPG Training.**   SA-DDPG uses the same hyperparameters as in DDPG training. For the additional regularization parameter $\kappa$ for $\pi(s)$, we choose in $\{0.1, 0.3, 1.0, 3.0\}$ for InvertedPendulum and Reacher due to their low dimensionality and $\{30, 100, 300, 1000\}$ for other environments.. We train the actor network without state-adversarial regularization for the first $1 \times 10^6$ steps, then increase $\epsilon_t$ from 0 to the target value in $5 \times 10^5$ steps, and then keep training at the target $\epsilon$ for $5 \times 10^5$ steps. The same $\epsilon$ schedule is used for both SGLD and convex relaxation. For SGLD, we run 5 iterations with step size $\frac{\epsilon_t}{5}$ and set the temperature parameter $\beta = 1 \times 10^{-5}$. For convex relaxations, we use the efficient IBP+Backward scheme [83], and a training schedule similar to [88] by mixing the IBP bounds and backward mode perturbation analysis bounds. The total number of training steps is thus $2 \times 10^6$, which is the same as the regular DDPG training. The target $\epsilon$ values for each task is the same as $\epsilon$ listed in Table 2 for evaluation.

Note that we apply perturbation on normalized environment states. The normalization factors are the standard deviations calculated using data collected on the baseline policy (vanilla DDPG) without adversaries.

## H    Additional Details for SA-DQN

**Algorithm**    We present the SA-DQN training algorithm in Algorithm 7. The main difference between SA-DQN and DQN is the additional state-adversarial regularizer $\mathcal{R}_{\text{DQN}}(\theta)$, which encourages the network not to change its output under perturbations on the state observation. We highlighted these changes in Algorithm 7. Note that the use of hinge loss is not required; other loss functions (e.g., cross-entropy loss) may also be used.

---

**Algorithm 7** State-Adversarial Deep Q-Learning (SA-DQN)

---

1:  Initialize current Q network $Q(s, a)$ with parameters $\theta$.
2:  Initialize target Q network $Q'(s, a)$ with parameters $\theta' \leftarrow \theta$.
3:  Initial replay buffer $\mathcal{B}$
4:  **for** $t = 1$ to $T$ **do**
5:      With probability $\epsilon_t$ select a random action at $a_t$, otherwise select $a_t = \arg\max_a Q_\theta(s_t, a; \theta)$
6:      Execute action $a_t$ in environment and observe reward $r_t$ and state $s_{t+1}$
7:      Store transition $\{s_t, a_t, r_t, s_{t+1}\}$ in $\mathcal{B}$.
8:      Randomly sample a minibatch of $N$ samples $\{s_i, a_i, r_i, s'_i\}$ from $\mathcal{B}$.
9:      For all $s_i$, compute $a_i^* = \arg\max_a Q_\theta(s_i, a; \theta)$.
10:     Set $y_i = r_i + \gamma \max_{a'} Q'_{\theta'}(s'_i, a'; \theta)$ for non-terminal $s_i$, and $y_i = r_i$ for terminal $s_i$.
11:     Compute TD-loss for each transition: TD-$L(s_i, a_i, s'_i; \theta) = \text{Huber}(y_i - Q_\theta(s_i, a_i; \theta))$
12:     Define $\mathcal{R}_{\text{DQN}}(\theta) := \sum_i \max\big\{ \max_{\hat{s}_i \in B(s)} \max_{a \neq a_i^*} Q_\theta(\hat{s}_i, a; \theta) - Q_\theta(\hat{s}_i, a_i^*; \theta), -c \big\}$.
13:     Option 1: Use projected gradient descent (PGD) to solve $\mathcal{R}_{\text{DQN}}(\theta)$.
14:         Run PGD to solve: $\hat{s}_i = \arg\max_{\hat{s}_i \in B(s_i)} \max_{a \neq a_i^*} Q_\theta(\hat{s}_i, a; \theta) - Q_\theta(\hat{s}_i, a_i^*; \theta)$.
15:         Compute the sum of hinge loss of each $s_i$:
            $\overline{\mathcal{R}}_{\text{DQN}}(\theta) = \sum_i \max\{\max_{a \neq a_i^*} Q_\theta(\hat{s}_i, a; \theta) - Q_\theta(\hat{s}_i, a_i^*), -c\}$.
16:     Option 2: Use convex relaxations of neural networks to solve a surrogate loss of $\mathcal{R}_{\text{DQN}}(\theta)$.
17:         For all $s_i$ and all $a \neq a_i^*$, obtain upper bounds on $Q_\theta(s, a; \theta) - Q_\theta(s, a_i^*; \theta)$:
            $u_{a_i^*, a}(s_i; \theta) = \text{ConvexRelaxUB}(Q_\theta(s, a; \theta) - Q_\theta(s, a_i^*; \theta), \theta, s \in B(s_i))$
18:         Compute a surrogate loss for the hinge loss:
            $\overline{\mathcal{R}}_{\text{DQN}}(\theta) = \sum_i \max\big\{ \max_{a \neq a_i^*}\{u_{a_i^*, a}(s_i)\}, -c \big\}$
19:     Perform a gradient descent step to minimize $\frac{1}{N}[\sum_i \text{TD-}L(s_i, a_i, s'_i; \theta) + \kappa_{\text{DQN}} \overline{\mathcal{R}}_{\text{DQN}}(\theta)]$.
20:     Update Target Network every $M$ steps: $\theta' \leftarrow \theta$.
21: **end for**

---

**Hyperparameters for Vanilla DQN training.**    For Atari games, the deep Q networks have 3 CNN layers followed by 2 fully connected layers (following [77]). The first CNN layer has 32 channels, a kernel size of 8, and stride 4. The second CNN layer has 64 channels, a kernel size of 4, and stride 2. The third CNN layer has 64 channels, a kernel size of 3, and stride 1. The fully connected layers have 512 hidden neurons for both value and advantage heads. We run each environment for $6 \times 10^6$ steps without framestack. We set learning rate as $6.25 \times 10^{-5}$ (following [26]) for Pong, Freeway and RoadRunner; for BankHeist our implementation cannot reliably converge within 6 million steps, so we reduce learning rate to $1 \times 10^{-5}$. For all Atari environments, we clip reward to $-1, +1$ (following [46]) and use a replay buffer with a capacity of $2 \times 10^5$.

We set discount factor set to 0.99. Prioritized replay buffer sampling is used with $\alpha = 0.5$ and $\beta$ increased from 0.4 to 1 linearly through the end of training. A batch size of 32 is used in training. Same as in [46], we choose Huber loss as the TD-loss. We update the target network every 2k steps for all environments.

**Hyperparameters for SA-DQN training.**    SA-DQN uses the same network structure and hyperparameters as in DQN training. The total number of SA-DQN training steps in all environments are the same as those in DQN (6 million). We update the target network every 2k steps for all

(a) Natural rewards (no attacks)　　　　　(b) Rewards under the best (strongest) attacks

Figure 9: Box plots of natural rewards and rewards under the strongest (best) attacks for PPO, adversarially trained PPO and SA-PPO agents corresponding to the results presented in Table 1 (Table 1 only reports mean and standard deviation). Each box shows the distribution of cumulated rewards collected from 50 episodes of a single agent. The red lines inside the boxes are median rewards, and the upper and lower sides of the boxes show 25% and 75% percentile rewards of 50 episodes. The line segments outside of the boxes show min or max rewards.

environments except that the target network is updated every 32k steps for RoadRunner's SA-DQN, which improves convergence for our short training schedule of 6 million frames. For the additional state-adversarial regularization parameter $\kappa$ for robustness, we choose $\kappa \in \{0.005, 0.01, 0.02\}$. For all 4 Atari environments, we train the Q network without regularization for the first $1.5 \times 10^6$ steps, then increase $\epsilon$ from 0 to the target value in $4 \times 10^6$ steps, and then keep training at the target $\epsilon$ for the rest $5 \times 10^5$ steps.

**Training Time** As Atari training is expensive, we train DQN and SA-DQN only 6 million frames; the rewards reported in most DQN paper (e.g., [46, 77, 26]) are obtained by training 20 million frames. Thus, the rewards (without attacks) reported maybe lower than some baselines. The training time for vanilla DQN, SA-DQN (SGLD) and SA-DQN (convex) are roughly 15 hours, 40 hours and 50 hours on a single 1080 Ti GPU, respectively. The training time of each environment varies but is very close.

Note that the training time for convex relaxation based method can be further reduced when using an more efficient relaxation. The fastest relaxation is interval bound propagation (IBP), however it is too inaccurate and can make training unstable and hard to tune [88]. We use the tighter IBP+Backward relaxation, and its complexity can be further improved to the same level as IBP with the recently developed loss fusion technique [83], while providing a much better relaxation than IBP. Our work simply uses convex relaxations as a blackbox tool and we leave further improvements on convex relaxation based methods as a future work.

# I  Additional Experimental Results

## I.1  More results on SA-PPO

**Box plots of rewards for SA-PPO agents** In Table 1, we report the mean and standard deviation of rewards for agents under attack. However, since the distribution of cumulative rewards can be non-Gaussian, in this section we include box plots of rewards for each task in Figure 9. We can observe that the rewards (median, 25% and 75% percentiles) under the strongest attacks (Figure 9b) significantly improve.

**Evaluation using multiple $\epsilon$** In Figure 10 we show the attack rewards of PPO and SA-PPO agents with different perturbation budget $\epsilon$. We can see that the lowest attack rewards of SA-PPO agents are higher than those of PPO under all $\epsilon$ values. Additionally, Robust Sarsa (RS) attacks and RS+MAD attacks are typically stronger than other attacks. On vanilla PPO agents, the MAD attack is also competitive.

**Convergence of PPO and SA-PPO agents** We want to confirm that our better performing Humanoid agents under state-adversarial regularization are not just by chance. We train each environment

Figure 10: Attacking PPO agents under different $\epsilon$ values. Each data point reported in this figure is an average of 50 episodes.

(a) Hopper

(b) Walker

(c) Humanoid

Figure 11: The median, 25% and 75% percentile episode reward of at least 15 PPO and 15 SA-PPO agents during training. The region of the shaded colors (light blue: SA-PPO solved with SGLD; light green: SA-PPO solved with convex relaxations; light red: vanilla PPO) represent the interval between 25% and 75% percentile rewards over these 15 different training runs, and the solid line is the median rewards over these runs.

using SA-PPO and PPO *at least 15 times*, and collect rewards during training. We plot the median, 25% and 75% percentile of rewards during the training process for all these runs in Figure 11.

We can see that our SA-PPO agents consistently outperform vanilla PPO agents in Humanoid. Since we also present the 25% and 75% percentile of the rewards among 15 agents, we believe this improvement is not because of cherry-picking. For Hopper and Walker environments, SA-PPO has almost no performance drop compared to vanilla PPO.

(a) Natural Rewards (no attacks)      (b) Rewards under the best (strongest) attacks

Figure 12: Box plots of natural and attack rewards for DDPG and SA-DDPG. Each box is obtained from **11 agents** trained with the same parameters as the agents reported in Table 2 and tested for 50 episodes (each sample of the box is an average reward over 50 episodes). The red lines inside the boxes are median rewards, and the upper and lower sides of the boxes show 25% and 75% percentile rewards. The line segments outside of the boxes show min or max rewards.

## I.2   More results on SA-DDPG

**Reproducibility over multiple training runs.** To show that our SA-DDPG can consistently obtain a robust agent and we do not cherry-pick good results, we repeatedly train all 5 environments using SA-DDPG and DDPG **11 times** each and attack all agents. We report the median, minimum, 25% and 75% rewards of 11 agents in box plots. The results are shown in Figure 12. We can observe that SA-DDPG is able to consistently improve the robustness: the median, 25% and 75% percentile rewards under attacks are significantly and consistently better than vanilla DDPG over all 5 environments.

**Full attack results** In Table 6 we present attack rewards on all of our DDPG agents. In the main text, we only report the strongest (lowest) attack rewards since the lowest reward determines the true agent robustness.

## I.3   Robustness Certificates

We report robustness certificates for SA-DQN in Table 3. As discussed in section E, for DQN we can guarantee that an action does not change under bounded adversarial noise. In Table 3, the "Action Cert. Rate" is the ratio of actions that does not change under any $\ell_\infty$ norm bounded noise. In some settings, we find that 100% of the actions are guaranteed to be unchanged (e.g., the Pong environment in Table 3). In that case, we can in fact also certify that the cumulative reward is not changed given the specific initial conditions for testing.

In SA-DDPG, we can obtain robustness certificates that give bounds on actions in the presence of bounded perturbation on state inputs. Given an input state $s$, we use convex relaxations of neural networks to obtain the upper and lower bounds for each action: $l_i(s) \leq \pi_i(\hat{s}) \leq u_i(s), \forall \hat{s} \in B(s)$. We consider the following certificates on $\pi(s)$: the average output range $\frac{\|u(s)-l(s)\|_1}{|\mathcal{A}|}$ which reflect the tightness of bounds, and the $\ell_2$ distance. Note that bounds on other $\ell_p$ norms can also be computed given $l_i(s)$ and $u_i(s)$. Since the action space is normalized within $[-1, 1]$, the worst case output range is 2. We report both certificates for all five environments in Table 7. DDPG without our robust regularizer usually cannot obtain non-vacuous certificates (range is close to 2). SA-DDPG can provide robustness certificates (bounded inputs guarantee bounded outputs). We include some discussions on these certificates in Section E.

For SA-PPO, since the action follows a Gaussian policy, we can upper bound its KL-divergence under state perturbations. The results are shown in Table 8. Note that, by increasing the regularization parameter $\kappa$, it is possible to obtain an even tighter certificate at the cost of model performance.

The robustness certificates for SA-DDPG and SA-PPO are computed using interval bound propagation (IBP). For vanilla DDPG and PPO, we use CROWN [86], a much tighter convex relaxation to obtain the certificates, but they are often still vacuous.

| Environment | | Ant | Hopper | Inverted Pendulum | Reacher | Walker2d |
|---|---|---|---|---|---|---|
| $\epsilon$ | | 0.2 | 0.075 | 0.3 | 1.5 | 0.05 |
| State Space | | 111 | 11 | 4 | 11 | 17 |
| Vanilla DDPG | Natural Reward | $1487 \pm 850$ | $3302 \pm 762$ | $1000 \pm 0$ | $-4.37 \pm 1.54$ | $1870 \pm 1418$ |
| | Critic Attack | $187 \pm 157$ | $2504 \pm 1207$ | $1000 \pm 0$ | $-24.35 \pm 5.10$ | $1301 \pm 1229$ |
| | Random Attack | $1473 \pm 795$ | $3086 \pm 1006$ | $1000 \pm 0$ | $-8.71 \pm 2.42$ | $1828 \pm 1456$ |
| | MAD Attack | $180 \pm 200$ | $2745 \pm 1073$ | $1000 \pm 0$ | $-27.67 \pm 5.32$ | $1564 \pm 1405$ |
| | RS Attack | $577 \pm 394$ | $606 \pm 124$ | $92 \pm 1$ | $-21.74 \pm 5.14$ | $959 \pm 1001$ |
| | RS+MAD | $112 \pm 165$ | $2056 \pm 1225$ | $1000 \pm 0$ | $-27.87 \pm 4.38$ | $790 \pm 985$ |
| | Best Attack | 112 | 606 | 92 | -27.87 | 790 |
| DDPG with adv. training (50% steps) Pattanaik et al. [50] | Natural Reward | $1522 \pm 831$ | $2694 \pm 497$ | $1000 \pm 0$ | $-5.20 \pm 1.70$ | $1818 \pm 1187$ |
| | Critic Attack | $222 \pm 299$ | $1789 \pm 1143$ | $703 \pm 373$ | $-23.88 \pm 5.05$ | $1391 \pm 1083$ |
| | Random Attack | $1389 \pm 785$ | $2316 \pm 741$ | $1000 \pm 0$ | $-9.09 \pm 2.42$ | $1793 \pm 955$ |
| | MAD Attack | $92 \pm 240$ | $1497 \pm 839$ | $238 \pm 240$ | $-25.81 \pm 6.53$ | $1680 \pm 1106$ |
| | RS Attack | $180 \pm 157$ | $41 \pm 105$ | $39 \pm 0$ | $-25.45 \pm 6.70$ | $837 \pm 722$ |
| | RS+MAD | $74 \pm 177$ | $1503 \pm 851$ | $116 \pm 90$ | $-25.81 \pm 6.53$ | $1120 \pm 859$ |
| | Best Attack | 74 | 41 | 39 | -25.81 | 837 |
| DDPG with adv. training (100% steps) Pattanaik et al. [50] | Natural Reward | $1082 \pm 574$ | $973 \pm 0$ | $1000 \pm 0$ | $-5.71 \pm 1.80$ | $462 \pm 569$ |
| | Critic Attack | $126 \pm 148$ | $62 \pm 34$ | $174 \pm 66$ | $-21.91 \pm 3.52$ | $809 \pm 525$ |
| | Random Attack | $832 \pm 545$ | $577 \pm 431$ | $998 \pm 5$ | $-9.60 \pm 2.56$ | $751 \pm 568$ |
| | MAD Attack | $43 \pm 165$ | $56 \pm 50$ | $121 \pm 19$ | $-26.47 \pm 4.19$ | $699 \pm 484$ |
| | RS Attack | $98 \pm 313$ | $24 \pm 15$ | $82 \pm 0$ | $-22.17 \pm 4.46$ | $302 \pm 260$ |
| | RS+MAD | $21 \pm 141$ | $56 \pm 50$ | $110 \pm 26$ | $-27.44 \pm 4.05$ | $488 \pm 406$ |
| | Best Attack | 21 | 24 | 82 | $-27.44$ | 302 |
| SA-DDPG solved by SGLD | Natural Reward | $2186 \pm 534$ | $3068 \pm 223$ | $1000 \pm 0$ | $-5 \pm 1$ | $3318 \pm 680$ |
| | Critic Attack | $2076 \pm 556$ | $2899 \pm 439$ | $423 \pm 281$ | $-12.10 \pm 4.58$ | $1210 \pm 979$ |
| | Random Attack | $2162 \pm 524$ | $3071 \pm 196$ | $1000 \pm 0$ | $-11.41 \pm 4.96$ | $3058 \pm 848$ |
| | MAD Attack | $2128 \pm 482$ | $3093 \pm 17$ | $733 \pm 284$ | $-11.94 \pm 4.79$ | $3252 \pm 689$ |
| | RS Attack | $1838 \pm 665$ | $1729 \pm 792$ | $832 \pm 328$ | $-11.69 \pm 4.80$ | $2224 \pm 1050$ |
| | RS+MAD | $2046 \pm 670$ | $1609 \pm 676$ | $724 \pm 322$ | $-12.01 \pm 4.84$ | $1933 \pm 1055$ |
| | Best Attack | **1838** | **1609** | 423 | **$-12.10$** | 1210 |
| SA-DDPG solved by convex relaxations | Natural Reward | $2254 \pm 430$ | $3128 \pm 453$ | $1000 \pm 0$ | $-5.24 \pm 2.06$ | $4540 \pm 1562$ |
| | Critic Attack | $1826 \pm 568$ | $2546 \pm 843$ | $1000 \pm 0$ | $-11.51 \pm 3.80$ | $2245 \pm 1881$ |
| | Random Attack | $2249 \pm 491$ | $3036 \pm 593$ | $1000 \pm 0$ | $-9.87 \pm 3.95$ | $4216 \pm 1616$ |
| | MAD Attack | $2106 \pm 573$ | $2959 \pm 663$ | $1000 \pm 0$ | $-12.43 \pm 3.76$ | $4135 \pm 1884$ |
| | RS Attack | $2276 \pm 242$ | $1258 \pm 561$ | $1000 \pm 0$ | $-11.40 \pm 3.56$ | $1986 \pm 1993$ |
| | RS+MAD | $2012 \pm 701$ | $1202 \pm 402$ | $1000 \pm 0$ | $-12.44 \pm 3.77$ | $2315 \pm 2127$ |
| | Best Attack | 1826 | 1202 | **1000** | $-12.44$ | **1986** |

Table 6: Rewards on 5 MuJoCo environments using policies trained by DDPG and SA-DDPG. Natural reward is the reward in clean environment without adversarial attacks. The "Best Attack" rows report the lowest reward over all five attacks (representing the strongest attack), and this lowest reward is used for robustness evaluation.

Table 7: Robustness certificates on bounded action changes under bounded state perturbations for DDPG agents. Results are averaged over 50 episodes. A smaller number is better. A vanilla DDPG agent typically cannot provide non-vacuous robustness guarantees.

| Settings | | Ant | Hopper | InvertedPendulum | Reacher | Walker2d |
|---|---|---|---|---|---|---|
| Certificates ($\ell_2$ upper bound) | SA-DDPG (Convex) | 0.181 | 0.050 | 0.787 | 0.202 | 0.169 |
| | DDPG (vanilla) | 3.972 | 2.612 | 0.992 | 1.491 | 2.484 |
| Certificates ($\ell_1$ upper bound) | SA-DDPG (Convex) | 0.454 | 0.074 | 0.787 | 0.283 | 0.301 |
| | DDPG (vanilla) | 11.087 | 4.345 | 0.992 | 2.107 | 4.923 |
| Certificates ($\ell_\infty$ upper bound) | SA-DDPG (Convex) | 0.104 | 0.041 | 0.787 | 0.157 | 0.131 |
| | DDPG (vanilla) | 1.734 | 1.794 | 0.992 | 1.073 | 1.570 |
| Certificates (Range) | SA-DDPG (Convex) | 0.057 | 0.025 | 0.787 | 0.142 | 0.050 |
| | DDPG (vanilla) | 1.386 | 1.448 | 0.992 | 1.054 | 0.821 |

Table 8: Upper bound on KL-divergence $D_{\mathrm{KL}}(\pi(a|s)\|\pi(a|\hat{s}))$ for three PPO environments. A smaller number is better. SA-PPO can reduce this upper bound significantly especially for high dimensional environments like Humanoid.

| Settings | | Hopper | Walker2d | Humanoid |
|---|---|---|---|---|
| Certificates (KL upper bound) | SA-PPO (Convex) | 0.1232 | 0.09831 | 3.529 |
| | PPO (vanilla) | 32.16 | 31.56 | 925140 |