[Reviews · NeurIPS 2020]

Review 1

Summary and Contributions: The authors describe a framework to study adversarial observation perturbations against Deep RL agents. To this, they define the state-adversarial Markov decision process (SA-MDP) as a formal and theoretical framework. In essence, the authors propose a policy regularization that enables an agent to approximate an original policy under attack from an optimal adversary, and give approximations of this method for PPO, DDPG, and DQN. They propose two novel attack methods against RL agents. The authors show experimental results of their policy regularization method for different algorithms and attacks.

Strengths: - The paper is well written and the reasoning seems sound - The general experimental design is sound, and covers all immediate cases. - The direction of research is relevant and the contributions are significant.

Weaknesses: The paper is too long (important aspects have been moved to the appendix, the conclusion rephrased as the broader impact section put of 9th page; broader impact section itself is missing). The paper is not self-contained. Experimental design: the presentation of the experimental results is not clear in general. More points: - the authors don’t give a std deviation across different training runs. This would be interesting to judge whether the small improvements for some models wrt. performance or robustness are significant. - The adversary bounds chosen are not explained or varied. - The baseline adversarial methods Critic and Random sometimes improve agent performance? - The adversaries implemented are quite weak. (Bit of speculation on my part): As far as I understand it, the regularization essentially constrains the policy to be smooth over the observation. Are the proposed algorithms (and rather heavy optimization methods) the right way to achieve this objective? The authors propose to use expensive second-order optimization for their method. I suppose the proposed method produces a huge computational workload. The authors miss to give an analysis on this in the experimental section.

Correctness: In general, the claims stated by the authors are reasonable and seem correct to me. They are supported by the experimental results.

Clarity: *** Derivations in Section 3 *** While the theorems across Section 3.1 seem reasonable I would have liked some a more self-contained presentation of theorems together with proofs. P4, 150-171 quickly jump over theorems without giving at least rough sketches of the proofs. Assumption 2 (Bounded adversary power) is a bit strange, and while the experimental implementation (with the norm ball around s) seems reasonable for many environments, this should probably be defined in a better way. The authors refer to the Appendix a lot and in my opinion such derivations are necessary for the reader to follow along. - However, I was curious about Theorem 3 – a proof is missing. - Moreover, Theorem 5 was left too unexplained. I cannot really follow how the authors get there. - How do you arrive at Theorem 6? Theoretical Results: *** Experiments in Section 4 *** I see a few issues with the (presentation of) the experimental results.: - apart from Figure 2, the experiment results are presented in dense tables without a clear way to judge the significance of the results. Add Plots (similar to Appendix I, Figure 12). - Table 3, gray markings: These are applied inconsistently. For Pong, SA-DQN PGD and convex, both should be marked (both 21.0). Also, I would like to take note that Vanilla DQN seems to perform comparably, within limits of the evaluation, with the author’s models. - Although it presents essentially the same type of results data, Table 1 employs a different structure than Tables 2 and 3. - Table 2 does not report which attack was the strongest. Given that against PPO, the novel RS attack massively outperforms the baseline attack methods, this would be interesting to see. - Across the board, adversarial training (adv. 50 / 100%) performs very bad. To save space, these results could be omitted. - Numbers (decimal points) are not aligned, which make It even harder to quickly parse the results. Moreover, I think the authors should add a few more experiments: - A runtime assessment of their method; 2nd order optimization sound computationally expensive. It would be great to see this in comparison to the performance of the baseline methods. - It would be interesting to the see an ablation study that analyzes the perturbation budget over the robustness of the policy. - I think the authors should also add a paragraph that lines out limitations of their method.

Relation to Prior Work: Yes.

Reproducibility: Yes

Additional Feedback: The abstract states that how to improve the robustness of DRL under adversarial settings has not been well studied. I think this claim is too strong as this has been subject to research in a lot of work. (Starting from line 138:) Why do you put $\tilde{V}^{\pi}_{\nu}$ ? I would rather use $V^{\pi \circ \nu}$ . This would seem more natural to me. Same for $Q$ Line 198: “neuron networks”  “neural networks” Line 204: \texttt{auto_LiRPA} is not well-known, at least not to me. Line 255+ and 267+ (the equations): please do not reduce font sizes. Add label. Lines 281-286 contains important information that cannot be moved to the appendix. ####################### Update ####################### I would like to thank the authors for their feedback. Indeed, your feedback together with the other reviews clarified a lot (and I got a few things a bit different, especially details in the experimental section). Most of my confusion came from the denseness of the paper and the missing running example that R2 proposes to add in. The authors also line out a few suggestions to improve. With such improvements I think the paper is much more readable and becomes a good fit to the conference.


Review 2

Summary and Contributions: The authors formulate a state-adversarial MDP, where an adversary can change the agent's perception to be a nearby state. They show basic results of policy optimization (including the existence of SA-MDPs where there is no optimal Markovian policy. They show a bound on policy performance and use it to define a policy regularizer to defend against attacks. In effect, their regularizer enforces that the policy action not change too much over a set of neighboring states. They run experiments on their approach and show that their approach improves performance in adversarial settings and some non-adversarial settings.

Strengths: The ideas behind the work are clean and potentially highly influential. Adversarial ML is a huge area of interest, and while adversarial RL has gotten some attention there is not a clear uniting formalism. I could see SA-MDPs filling that gap. The methods proposed look sound, and the policy regularizer seems like it may be useful in non-adversarial settings by enforcing a type of smoothness prior on the policy.

Weaknesses: The idea behind the work is good, and the results presented look sound. The primary weakness of the paper is that it is quite dense and detail oriented. See the section on clarity for details.

Correctness: The claims and methods look correct and the empirical methodology is appropriate.

Clarity: The paper is clear, but somewhat hard to follow. There's room for improvement to make it more engaging and easier to follow. In particular, especially when introducing a new framework, it is helpful to include a running example and provide intuitive groundings of the theorems. For example, consider describing the impact of the policy regularizer on a toy problem and/or walking through an example (Fig. 1 tries to do this, but isn't connect to the SA-MDP formalism direction). The paper often leaves important conclusions unstated for the reader to infer. The clearest example of this is on the bottom of page 4. The paper presents several theorems that are described in the text as negative results, but the only statement of the results is the theorem itself. Thms 3-5 should be described informally in English concisely in addition to the formal theorem statement. I also think it is important to describe something of the proof here --- it is ok to move detailed proofs to the appendix, but the paper should stand on its own. In this case, I think that means including an example or proof sketch in the main text. This also shows up in that the paper does not have a discussion or conclusion section to interpret the experimental results. I realize that a lot of these choices are driven by space constraints. I recommend that the authors consider the most important parts of the paper in order to tell a full story and then move additional sections to the appendix. For example, instead of covering everything briefly (e.g., all of the robust optimization approaches and adversarial attacks) consider covering one in detail, the others briefly, and then referring readers to the appendix.

Relation to Prior Work: Prior work is discussed very well, but the paper does not do a good job of describing the relationship between this work and prior work. The related work section is a good literature review, but does not situate these contributions with respect to that work.

Reproducibility: Yes

Additional Feedback: Typos: I am curious if there are any environments (or types of optimal policy) where this method would fail. For example, if the policy really does need to change sharply in a particular neighborhood, then it seems like this would substantially harm natural performance. Can the authors discuss this somewhat? Can the authors comment more on the choice to sample states from B(s), rather than doing a maximization. I understand that this may be useful, but the properties of the max may be quite different than the sum. l.137 "to those of regular MDP" l.152 "the known results in MDP" --- while the literal acronym works, this reads strangely to me. Consider 'known results in MDP theory' l.163 "not all hopes are lost" --- phrase is 'not all hope is lost' but consider replacing with something simpler Update after the discussion phase: I appreciate the author's response and the description of limitations they included. I am not changing my (quite high) evaluation.


Review 3

Summary and Contributions: Existing techniques for improving robustness of policies are ineffective for many RL tasks, so they propose a new formulation of MDPs called state-adversarial Markov decision process (SA-MDP). They show that while there does not exist an optimal policy under an optimal adversary in this SA-MDP formulation (as does in classical MDPs), the loss in performance can be bounded under certain assumptions. Under this formulation, they develop a policy regularization technique that is robust to noise and adversarial attacks on state observations and improves the robustness of DQN ,PPO, and DDPG in both discrete and continuous spaces. The authors further evaluate their algorithm under 2 new attacks, the robust SARSA attack and the maximal action difference (MAD) attack. They show that their method outperforms several baselines on 11 environments in both the discrete and continuous action space to prove their results.

Strengths: - With more robust RL, we can more safely apply RL in real world settings, such as autonomous driving and situations where small amounts of noise can cause huge consequences. One weakness of deep RL is that it often fails to generalize at test time, and with a more robust method, perhaps RL will be able to be used in other applications where only other methods are currently used. - The authors give a clear and easy to follow presentation of theorems which lead to the motivation behind the policy regularization technique. - Experiments are consistent with their claims and do show that their SA-DQN outperforms several baselines against the strongest adversaries - RL robustness is very relevant in the realistic settings mentioned, including sensor noise and measurement errors - Train humanoid more robustly, which is great!

Weaknesses: - In the related work section: Adversarial Attacks on State Observations in DRL, would be nice to know how these attacks relate to the ones they looked at (did they use these or what’s different about the attacks they used) - It might be interesting to evaluate why the agent performs worse under the robust SARSA than under the MAD attack, since the MAD attack is designed to maximize exactly what the proposed regularizer tries to minimize - small typo in line 810 of appendix - Figure 1 is shown, but no experiments on this environment. Also, safeness is mentioned throughout the introduction, but none of the experiments are specified that safeness is critical (i.e. non-resetting the humanoid / ant when it falls over, other cliff falling states / disastrous outcome states) Other minor comments: - Unclear difference between partial observability and state adversarial. - Would be interesting to see how practical implementations of TD3 or SAC, which use target Q functions / other hacks for stability, are affected by the theory. It could be possible that the noisiness of state estimation is already accounted for in these hacks, so having more robust policies shows no significant improvement. - Model based methods? If this does not work on something like MPC, then it’s possible that it is merely attacking the networks parameterizing the policies, rather than the some inherit robustness to attacks on state perturbation. - Table 1 would be nicer if it was flush bottom, rather than having a small paragraph of text between the table and the bottom of the page.

Correctness: Nothing pops out as incorrect (but I also didn’t read through the proofs in the appendix very thoroughly)

Clarity: Paper is structured very clearly and is easy to follow. The theorems provide motivation for why minimizing the variation distance between the policies under two MDP formulations will make the policy more robust, and follows with examples in 3 widely used RL algorithms.

Relation to Prior Work: Overall, looks good.

Reproducibility: Yes

Additional Feedback: I have read the author response and the reviews from other authors. I'm satisfied with the response and stick to my original score of Accept.


Review 4

Summary and Contributions: This paper tackles adversarial perturbations on state observations in RL. It describes an approach to train robust policies with a modified loss for standard RL algorithms such as ddpg e.g. The main idea consists in smoothing the randomized policy, in such a way that for each state, in the neighborhood of that state, the distribution of actions does not change too much.

Strengths: - The problem adressed in the paper (attacks on state observations) is highly relevant for NeurIps and very interesting. - The method proposed in the paper (a regularizer entailing smoothness) is sound - The method is implemented in several algorithms - The theory part is sound. In particular, theorem 5 relates the true objective (robustness) to the regularizer nicely.

Weaknesses: - The idea of the paper is that smooth policies are also robust policies. In the supervised setting, this fact is well known, well understood, and many smoothing techniques are available, including the one presented in this paper [1]. The paper would have been more interesting if many smoothing techniques were compared in the RL setting.

Correctness: Theorems are correct as far as I can tell, although their proofs are probably lengthier than needed.

Clarity: The paper is well written.

Relation to Prior Work: The authors should refer to the litterature on smoothing techniques for defending against adversarial attacks in the supervised learning setting. In particular, virtual adversarial training (VAT), described in [1] is exactly the same regularizer as described in this paper, and should absolutely be cited. The same idea, although a little different, is presented in [2]. [1] T Miyato, S. Maeda, M. Koyama, K. Nakae, S. Ishii, Distributional Smoothing with Virtual Adversarial Training [2] H. Zhang, Y.Yu, J.Jiao, E.Xing, L.E.Ghaoui, M.Jordan, Theoretically Principled Trade-off between Robustness and Accuracy

Reproducibility: Yes

Additional Feedback: - the fact that optimal policies are randomized is not particularly surprising here, because we are in a zero-sum two player game, where randomizing improves the value of the game in general. - because \tilde{V}_{\nu}^{\pi}=V^{\nu\circ\pi}, theorem 1 is absolutely trivial. - Theorem 5, in the definition of \alpha, is the \gamma/(1-\gamma)^2 term really tight ? Can't we replace it by \gamma/(1-\gamma) ?

[Author Response · NeurIPS 2020]

**General response.** Reviewers give great feedback on improving the structure of this paper under space constraints, and we plan to reorganize our paper: (1) Move non-critical theorems and optimization techniques to appendix and leave space for discussions and proof sketches. (2) Include a small running example (as in Appendix A) of SA-MDP. (3) Rephrase any claims that seem too strong, add additional reference and discuss more connections to previous works. (4) Use more plots (like Fig. 11 and 12) (5) Fix typos, format and refine notations.

**R1.** Paper too long We will reorganize our paper (see general response). Std. across training runs In Fig. 11 and 12 (appendix), the rewards are collected from 30 and 11 training runs for PPO and DDPG, respectively. In **Table B**, we train DQN and SA-DQN >5 times. The red lines in bars represent median rewards. We improve reward under attacks consistently across runs. Adversary bounds We use the $\ell_\infty$ norm based adversary bounds as in many works on attacking Deep RL [20,24,29,42,69]. We vary $\epsilon$ bounds in Fig. 9. Critic/Random attacks improve performance The small "improvement" in random attack is just by chance (Fig. 9 is more clear; yellow lines fluctuate). Critic attack sometimes improves PPO performance (green lines of Fig. 9). It is not a bug. In PPO, the critic is a value function $V(s)$ rather than $Q(s, a)$, thus critic attack is applied differently (appendix L676-681): the "attack" searchers a state with the worst value in $B(s)$, and the agent takes the action for the worst case. It is a more conservative action which sometimes prevents the agent from failing and improves performance. Weak adversaries implemented Our proposed robust Sarsa (RS) and MAD adversaries are not weak. From Table 1 and 7, our two new attacks are considerably stronger than the commonly used critic attack. 2nd-order optimization expensive We avoid 2nd order optimization (L180-181). SGLD (L188-196) is a first order method and only requires gradients. The convex relaxation method (L197-207) first produces a relaxed counterpart of the underlying neural network, then uses gradient descent to optimize it. Assumption 2 strange We need this assumption otherwise the adversary can arbitrarily change state and make the problem trivial. Practically it is a norm constraint as in [20,24,29,42,69]. Explain Thm 5 and 6 Following Thm 4 we cannot find an Markovian optimal policy for SA-MDP. Instead, Thm 5 upper bounds the performance loss by regularizing total variation (TV) distance. Thm 6 gives TV distance for DDPG. Thm 3 proof See appendix L616-620. Vanilla DQN performs comparably Vanilla DQN performs comparably only under clean evaluation; it performs poorly under attacks. For Pong, the reward is the lowest possible reward (-21). Table 2 structure and more results Full results for each attack are in appendix Table 7 to save space. Runtime assessment See **Table A**. Ablation study for perturbation budget In Fig. 9, we analyze the agent performance over different perturbation budgets $\epsilon$. Limitations See reply to R2.

**R2.** We will reorganize our paper as suggested, detailed in our general response. Limitations It is possible to construct an MDP that every nearby state requires a vastly different action, so a typical robustness prior does not hold. In the classification setting, a similar situation is to learn a parity function $f(x) = x_1 \oplus x_2 \cdots \oplus x_n$ ($\oplus$ is XOR) where robustness is impossible. For most realistic problems it's reasonable to assume that a robustness/smoothness prior is valid and helpful. Sum instead of max max represents the strongest adversary; sum or expectation over $B(s)$ is similar to adding random noise with certain distribution. This is a weaker adversary (like random attack in Table 1 and 7).

**R3.** Related attacks We will enhance the related work section as suggested. Existing attacks rely on the critic learned with the policy. Our MAD and RS attacks do not depend on this critic as using it can be suboptimal (L241-246). Why RS attack better than MAD MAD is myopic and maximizes one step difference without reducing cumulative rewards. RS attack learns a robust *action-value function*, where by definition gives a worst action to reduce cumulative rewards. Safeness specifications We conduct additional experiments on Ant and Humanoid and define the *safe rate* as the percentage that agent does not fall over 50 episodes. Vanilla DDPG (PPO) achieves 56% (2%) safe rate without attacks and 0% (0%) under attack, while SA-DDPG (SA-PPO) achieves 100% (68%) safe rate without attacks and 100% (34%) under attack for Ant (Humanoid, respectively). Partial observability In PO-MDPs, the observation is statistically related to groundtruth state. In SA-MDPs, the observation is an adversarially perturbed state: the adversary is assumed to know the weakness of the policy and can supply the worst-case state, which cannot be directly characterized as conditional observation probabilities in PO-MDP. SAC and TD3 We conduct experiments and find SAC policies are also not robust. SA-SAC significantly improves robustness (**Table C**). We leave model based methods as future work.

**R4.** Related work We will discuss the connection to smoothing in supervised learning and zero-sum game. We already cited Zhang et al. as [75] and will cite Miyato et al. For RL, not all techniques from supervised learning can be applied directly (line 32-36), so our theory is still valuable. Tighten constant Thank you for pointing this out. One $1/(1 - \gamma)$ factor in our bound is to cancel out a $(1 - \gamma)$ in the definition of $d_{s_0}^\pi$ in (20) in the appendix. Another $1/(1 - \gamma)$ factor is from the sum of a geometric sequence. We cannot see an obvious way to tighten it but will keep thinking about it.

**Table A:** Training time

| Method/Model | vanilla | SA (PGD/SGLD) | SA (Conv) |
|---|---|---|---|
| DDPG (Reacher) | 5.21h | 7.10h | 6.75h |
| DDPG (Ant) | 6.08h | 8.16h | 7.70h |
| PPO (Hopper) | 0.57h | 1.17h | 1.38h |
| PPO (Walker) | 0.61h | 1.56h | 1.80h |
| PPO (Humanoid) | 4.63h | 11.0h | 20.3h |
| DQN (RoadRunner) | 15.2h | 38.6h | 46.5h |
| DQN (Freeway) | 14.9h | 44.7h | 57.7h |

**Table B:** Box plot to show DQN performance with and without attacks across training runs. We train each setting at least 5 times (DQN training is expensive).

**Table C:** The median model performance of 11 training runs for SAC

| Env. | $\epsilon$ | Method | Natural Reward | Best Attack Reward |
|---|---|---|---|---|
| Hopper | .075 | SAC | $3494 \pm 3$ | $808 \pm 42$ |
| | | SA-SAC | $3553 \pm 7$ | $1478 \pm 220$ |
| Walker | .05 | SAC | $4371 \pm 39$ | $1725 \pm 1551$ |
| | | SA-SAC | $4126 \pm 80$ | $3854 \pm 109$ |
| Ant | .2 | SAC | $5236 \pm 628$ | $-212 \pm 348$ |
| | | SA-SAC | $4728 \pm 603$ | $1940 \pm 1612$ |

[Meta-Review · NeurIPS 2020]

The reviewers agreed that this is a strong, well-executed, and potentially high-impact work on a very timely topic. Congratulations! The reviewers also pointed out several presentation issues, and potential solutions. Please address these in the final version.